# YAP and TAZ regulate adherens junction dynamics and endothelial cell distribution during vascular development

Filipa Neto[1,2], Alexandra Klaus-Bergmann[1,3], Yu Ting Ong[4], Silvanus Alt[1], Anne-Clémence Vion[1,3], Anna Szymborska[1,3], Joana R Carvalho[5], Irene Hollfinger[1], Eireen Bartels-Klein[1,3], Claudio A Franco[5], Michael Potente[3,4,6]*, Holger Gerhardt[1,2,3,7,8,9]*

[1]Max-Delbrück-Center for Molecular Medicine, Berlin, Germany; [2]Vascular Biology Laboratory, Lincoln's Inn Fields Laboratories, London Research Institute – Cancer Research UK, London, United Kingdom; [3]DZHK (German Center for Cardiovascular Research), Berlin, Germany; [4]Angiogenesis and Metabolism Laboratory, Max Planck Institute for Heart and Lung Research, Bad Nauheim, Germany; [5]Vascular Morphogenesis Laboratory, Instituto de Medicina Molecular, Faculdade de Medicina da Universidade de Lisboa, Lisboa, Portugal; [6]International Institute of Molecular and Cell Biology, Warsaw, Poland; [7]Vascular Patterning Laboratory, Vesalius Research Center, Leuven, Belgium; [8]Department of Oncology, KU Leuven, Leuven, Belgium; [9]Berlin Institute of Health, Berlin, Germany

*For correspondence:
michael.potente@mpi-bn.mpg.de
(MP);
holger.gerhardt@mdc-berlin.de
(HG)

**Abstract** Formation of blood vessel networks by sprouting angiogenesis is critical for tissue growth, homeostasis and regeneration. How endothelial cells arise in adequate numbers and arrange suitably to shape functional vascular networks is poorly understood. Here we show that YAP/TAZ promote stretch-induced proliferation and rearrangements of endothelial cells whilst preventing bleeding in developing vessels. Mechanistically, YAP/TAZ increase the turnover of VE-Cadherin and the formation of junction associated intermediate lamellipodia, promoting both cell migration and barrier function maintenance. This is achieved in part by lowering BMP signalling. Consequently, the loss of YAP/TAZ in the mouse leads to stunted sprouting with local aggregation as well as scarcity of endothelial cells, branching irregularities and junction defects. Forced nuclear activity of TAZ instead drives hypersprouting and vascular hyperplasia. We propose a new model in which YAP/TAZ integrate mechanical signals with BMP signaling to maintain junctional compliance and integrity whilst balancing endothelial cell rearrangements in angiogenic vessels.
DOI: https://doi.org/10.7554/eLife.31037.001

## Introduction

A long-standing question in developmental and cell biology relates to how cells integrate mechanical and chemical signals to orchestrate the morphogenic behaviours that ensure adequate tissue patterning. During sprouting angiogenesis, the arrangement and distribution of cells rather than their numbers appear to drive morphogenesis of the vascular tree. Recent data showing unaltered remodelling in the absence of endothelial cell apoptosis and normal branching frequency across a range of endothelial cell densities support this idea (*Watson et al., 2016*). In the extreme, however, too few cells will jeopardize network formation and stability (*Phng et al., 2009*), whereas too many cells might compromise vessel calibre control (*Watson et al., 2016*). Functional network formation therefore needs to establish the right number of cells in the right place, and distribute them such that the

hierarchical branching pattern is supported. What establishes such a balance has remained unclear. Here we provide evidence for the yes-associated protein 1 (YAP) and its paralog WW domain containing transcription regulator 1 (TAZ) as critical endothelial cell autonomous regulators in this process.

YAP and TAZ, two transcriptional co-activators initially discovered as effectors of the Hippo signalling pathway, play a central role in organ size control via regulation of proliferation and apoptosis (*Piccolo et al., 2014*; *Meng et al., 2016*; *Yu et al., 2015*). In confluent cells, YAP and TAZ are phosphorylated by kinases of the Hippo pathway, which induces their retention in the cytoplasm. In sparse cells, YAP and TAZ can translocate to the nucleus, where they interact with transcription factors to regulate the expression of pro-proliferative and anti-apoptotic genes. Other stimuli have been found to regulate YAP and TAZ nuclear translocation and activity – these include, among others, G-protein coupled receptors (GPCRs) (*Yu et al., 2012*), junctional proteins (*Giampietro et al., 2015*; *Schlegelmilch et al., 2011*), and mechanical stimuli (*Dupont et al., 2011*; *Aragona et al., 2013*). Furthermore, besides cell proliferation and apoptosis, YAP and TAZ also regulate cell differentiation (*Yu and Guan, 2013*), migration (*Zhang et al., 2015*) and actomyosin contraction (*Lin et al., 2017*). In vascular development, the roles of YAP and TAZ are not fully understood. *Yap* null mutant zebrafish develop an initially normal vasculature but display increased vessel collapse and regression. *Yap/Taz* double mutant zebrafish die before the onset of circulation with severe developmental defects, precluding analysis of vascular development in this context (*Nakajima et al., 2017*). Endothelial-specific deletion of *Yap* in mice using the Tie2-Cre transgenic line is embryonically lethal due to heart valve defects caused by failed endothelial-to-mesenchymal transition (*Zhang et al., 2014*). During post-natal development of the mouse retina, YAP was shown to regulate vascular branching and density by promoting the transcription of *Angiopoetin-2* (16). While these studies point towards an important role for YAP in regulating blood vessel formation and maintenance, the cellular principles and effectors of YAP/TAZ in endothelial cells in vivo, as well as the possible interplay between YAP/TAZ and the major signalling pathways regulating angiogenesis remain poorly understood.

Here, we used loss and gain of function endothelial specific mouse models to address the roles of YAP and TAZ in the vasculature. We show that YAP and TAZ are both expressed and active in sprouting ECs and critical for sprouting angiogenesis. The inducible, endothelial-specific deletion of YAP and TAZ leads to severe morphogenic defects consistent with impaired junctional remodelling in vivo. We found that the loss of YAP and TAZ decreased VE-Cadherin turnover and decreased the frequency of junction associated intermediate lamellipodia. Furthermore, the loss of YAP and TAZ decreased cell migration and increased cell-cell coupling. We also discovered that endothelial YAP and TAZ strongly inhibit BMP signalling in vitro and in vivo, and that this is mechanistically linked to the migration and permeability defects. Together our results suggest that YAP and TAZ integrate mechanical stimuli with key transcriptional regulators of endothelial sprouting and cell rearrangements during angiogenesis.

## Results

### YAP and TAZ have distinct expression patterns in endothelial cells of developing vessels and localise to the nucleus at the sprouting front

Immunofluorescence staining in the postnatal mouse retina showed that YAP and TAZ are distinctly expressed in the ECs of the developing vasculature (*Figure 1*). While YAP is evenly expressed throughout the vasculature (*Figure 1A–D*), the expression of TAZ is especially prominent at the sprouting front (*Figure 1E–H*). Furthermore, YAP is exclusively cytoplasmic in all areas of the retinal vasculature, with the exception of the sprouting front where some ECs express nuclear YAP, although at lower levels than in the cytoplasm (*Figure 1A'–D'*). TAZ staining signal is very low in the remodelling plexus, arteries and veins (*Figure 1E'–H'*); at the sprouting front, TAZ is strongly nuclear in numerous ECs (*Figure 1E*, green arrowheads and E'), and both nuclear and cytoplasmic in others (*Figure 1E*, red arrowheads). The nuclear signal of YAP and TAZ did not correlate with a tip or stalk cell phenotype; nuclear YAP and TAZ are rather present in a subset of tip and stalk ECs at the sprouting front. YAP and TAZ were also found at endothelial adherens junctions in veins and in the remodelling plexus, (yellow arrowheads in *Figure 1D' and F'*), as revealed by co-staining for VE-

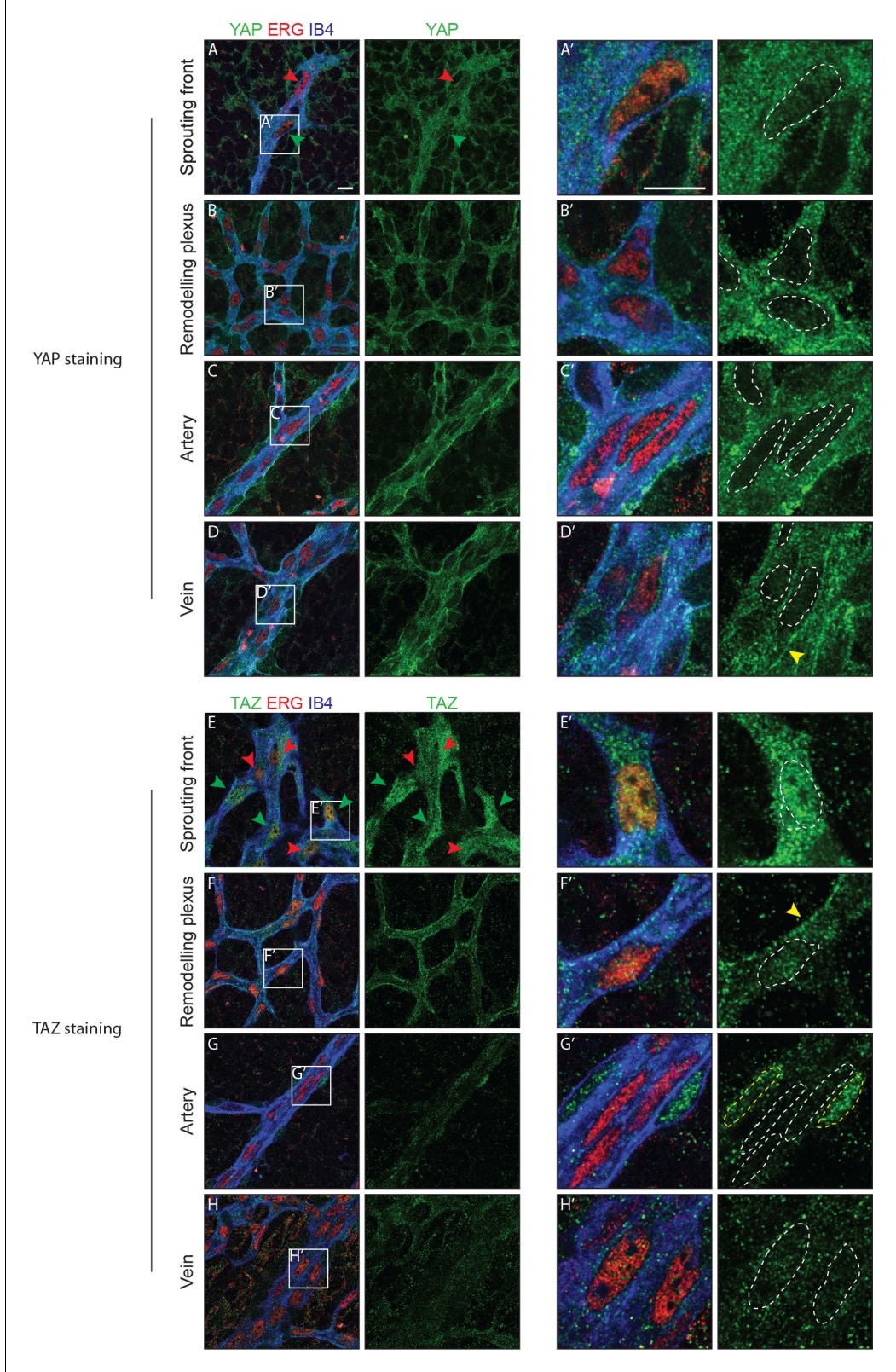

**Figure 1.** YAP and TAZ are expressed throughout the vasculature of developing mouse retinas, and localise to the nucleus of sprouting endothelial cells. Immunofluorescence staining of YAP (green, **A–D and A'–D'**) and TAZ (green, **E–H and E'–H'**) was performed in wild-type mouse retinas at post-natal day 6 (P6). Retinas were co-stained with the endothelial membrane marker Isolectin-B4 (IB4; blue) and with antibodies against the endothelial nuclei

*Figure 1 continued on next page*

*Figure 1 continued*

marker ERG (red). White dotted lines, outline of endothelial nuclei. Yellow dotted lines, outline of perivascular cells' nuclei. Green arrowheads, nuclear localisation of YAP and TAZ. Red arrowheads, cytoplasmic localisation of YAP and TAZ. Yellow arrowheads, junctional localisation of YAP and TAZ. Images correspond to single confocal planes. n > 3 animals for each staining. Scale bar: 10 µm.

DOI: https://doi.org/10.7554/eLife.31037.002

The following figure supplement is available for figure 1:

**Figure supplement 1.** YAP and TAZ localise at endothelial adherens junctions in the mouse retina.

DOI: https://doi.org/10.7554/eLife.31037.003

---

Cadherin (*Figure 1—figure supplement 1*). Together, these observations suggest that YAP/TAZ are abundant proteins in the endothelium, which are dynamically regulated during the angiogenic process.

## YAP/TAZ are required for vascular growth, branching and regularity of the network

To examine the cell-autonomous role of endothelial YAP and TAZ during angiogenesis we crossed mice bearing *floxed* alleles of *Yap* or *Taz* (*Gruber et al., 2016*) with mice expressing a tamoxifen-inducible Cre recombinase driven by the endothelial-restricted *Pdgfb* promoter (*Pdgfb-iCreERT2*) (*Claxton et al., 2008*). Injection of the offspring with tamoxifen induced loss of YAP and TAZ protein in ECs during post-natal vascular development, as evidenced by immunofluorescence staining (*Figure 2—figure supplement 1*).

Endothelial deletion of YAP or TAZ led to mild vascular defects (*Figure 2A,B,C,D*). $Yap^{fl/fl}$ *Pdgfb-iCreERT2* mice (*Yap* iEC-KO) presented reduced radial expansion of the vasculature (7% ± 5.4 reduction, p=0.0123) and reduced vessel density (9% ± 4.4 reduction, p=0.0002) (*Figure 2G,H*). $Taz^{fl/fl}$ *Pdgfb-iCreERT2* mice (*Taz* iEC-KO) did not show altered radial expansion but displayed decreased vessel density (6% ± 5.8 reduction, p=0.0214) (*Figure 2HG*). Neither mutant showed a change in the branching frequency of vessels (*Figure 2I*). Interestingly, in *Yap* iEC-KO retinas the expression of TAZ was increased and TAZ more often localised to the nucleus (*Figure 2—figure supplement 2*), suggesting compensatory regulation. *Taz* iEC-KO retinas did not however show a clear difference in YAP expression (data not shown). Deleting both proteins in compound mutant mice ($Yap^{fl/f}Taz^{fl/fl}$*Pdgfb-iCreERT2*, *YapTaz* iEC-KO) led to a dramatic defect in blood vessel development (*Figure 2E,F*): the retinal vasculature showed a 21% (±14, p=0.0012) decrease in radial expansion (*Figure 2G*), a 26% (±7.0, p<0.0001) decrease in capillary density (*Figure 2H*), and a 55% (±15.4, p<0.0001) decrease in branching frequency (*Figure 2I*). Interestingly, the vessel loops were not only bigger in *Yap/Taz* iEC-KO mice (*Figure 2J*), but also more variable in size (*Figure 2K*), and shape (*Figure 2L*) than in control mice. These results indicate that endothelial YAP and TAZ are critical for the development of a homogeneous blood vessel network and can perform redundant functions in the endothelium.

## YAP is required for endothelial cell proliferation in response to mechanical stretch

As YAP and TAZ display pro-proliferative and anti-apoptotic roles in many cell types (*Piccolo et al., 2014*; *Meng et al., 2016*), we evaluated whether the reduced vascularization of *Yap/Taz* iEC-KO retinas was associated with reduced cell proliferation or increased apoptosis. EC proliferation, assessed by EdU staining (*Figure 3A–C*), was decreased in *Yap* iEC-KO retinas (23% ± 10.0, p=0.0469), whilst not affected in *Taz* iEC-KO. Consistent with our prior results the decrease in cell proliferation was strongest in *Yap/Taz* iEC-KO retinas (33% ± 26.0, p=0.0059). Staining for cleaved caspase 3 revealed that apoptosis was unaffected by YAP/TAZ loss (*Figure 3D–F*).

To understand if YAP and TAZ were required for proliferation downstream of VEGF, we knocked down YAP and TAZ in human umbilical vein endothelial cells (HUVECs) using small interfering RNAs (siRNAs) (*Figure 3—figure supplement 1*) and measured the proliferation rate by flow cytometry after treatment with increasing concentrations of VEGF (*Figure 3G*). Interestingly, upon loss of YAP, TAZ or YAP/TAZ, ECs proliferated at similar or even increased rates compared to control cells.

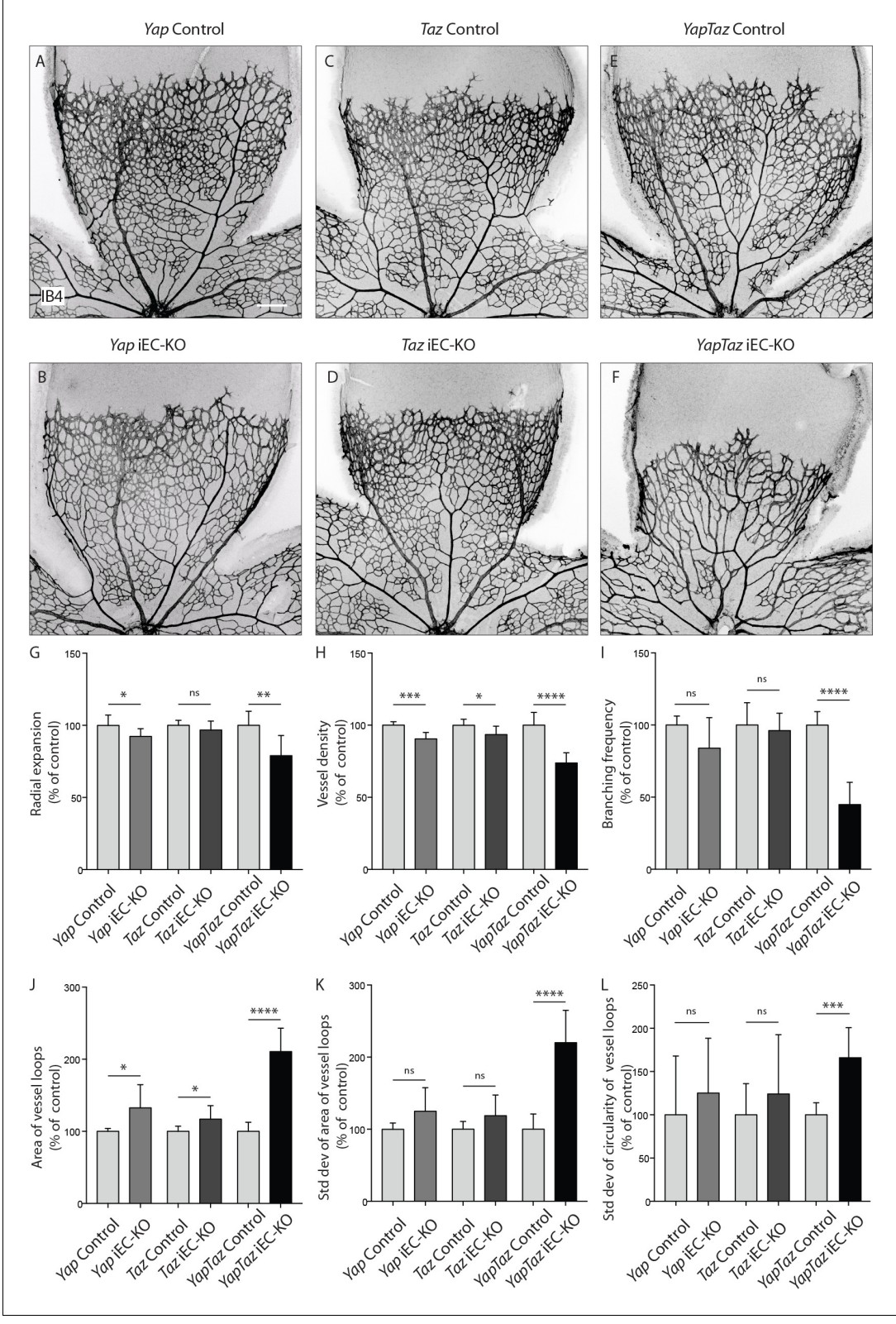

**Figure 2.** Endothelial YAP and TAZ are required for vessel growth, branching and homogeneity of the plexus. (A–F,) Retinas from P6 *Yap* iEC-KO (B), *Taz* iEC-KO (D) and *YapTaz* iEC-KO (F), and respective control pups (A,C,E) were stained with Isolectin B4 (IB4). Scale bar: 200 μm. (G–J), Quantification of radial expansion (G), vessel density (H), branching frequency (I) and area of vessel loops (J) in *Yap* iEC-KO, *Taz* iEC-KO and *YapTaz* iEC-KO. Results

*Figure 2 continued on next page*

*Figure 2 continued*

are shown as percentage of the respective controls. Data are mean ±SD. n ≥ 5 pups. *p* values were calculated using unpaired *t*-test. *p<0.05; **p<0.01; ****p<0.0001. (K, L), Quantification of the standard deviation of the area (K) and circularity (L) of the vessels loops in *Yap* iEC-KO, *Taz* iEC-KO and *YapTaz* iEC-KO retinas. Results are shown as percentage of the respective controls. Data are mean ±SD. n ≥ 5 pups. *p* values were calculated using unpaired *t*-test. *p<0.05; **p<0.01; ***p<0.001****p<0.0001.
DOI: https://doi.org/10.7554/eLife.31037.004

The following source data and figure supplements are available for figure 2:

**Source data 1.** Values for quantification of radial expansion (*Figure 2G*), vessel density (*Figure 2H*), branching frequency (*Figure 2I*), area of gaps (*Figure 2J*) and standard deviation of area (*Figure 2K*) and circularity (*Figure 2L*) of gaps in P6 *Yap* iEC-KO, *Taz* iEC-KO and *YapTaz* iEC-KO and respective control pups.
DOI: https://doi.org/10.7554/eLife.31037.007

**Figure supplement 1.** YAP and TAZ proteins are lost upon Cre-mediated genetic deletion in P6 mouse retinas.
DOI: https://doi.org/10.7554/eLife.31037.005

**Figure supplement 2.** TAZ compensates for the loss of YAP in endothelial cells in vivo.
DOI: https://doi.org/10.7554/eLife.31037.006

Furthermore, VEGF treatment did not alter the subcellular localisation of YAP and TAZ in HUVECs (*Figure 3—figure supplement 2*), suggesting that VEGF is not a primary regulator of their activity.

We next asked whether YAP and TAZ mediate endothelial proliferation in response to stretch – another crucial mitogenic stimulus for the endothelium (*Liu et al., 2007*). To this end, we subjected HUVECs to 24 hr of stretch and measured the proliferation rate in comparison to non-stretched, static cells treated with the same siRNAs, by EdU labelling (*Figure 3H*). Control cells responded to stretch with a 5-fold average increase in proliferation, and this effect was reduced upon knockdown of VE-Cadherin confirming previous observations (*Liu et al., 2007*). The knockdown of YAP, but not of TAZ, led to a significant decrease in stretch-induced proliferation. The knockdown of YAP/TAZ showed a tendency to decreased proliferation in response to stretch but did not reach statistical significance. Thus YAP is required for endothelial cell proliferation in response to mechanical stimulation at cell-cell junctions.

## YAP/TAZ loss leads to irregular endothelial cell distribution and haemorrhages

Further analysis of *Yap/Taz* iEC-KO retinas revealed severe defects at the sprouting front. *Yap/Taz* iEC-KO mutant retinas had 23% (±12.3, p=0.0113) fewer angiogenic sprouts than the control (*Figure 4A,B* yellow asterisks and *Figure 4—figure supplement 1*). Moreover, whereas control sprouts were elongated and showed long cellular protrusions towards the non-vascularised front (*Figure 4A'*), sprouts in *Yap/Taz* iEC-KO retinas were rounder and lacked protrusions (*Figure 4B'*). The defective sprout morphology correlated with irregular spacing and frequent aggregations of ECs within the sprouts (*Figure 4B'*), arguing that migration and/or the rearrangement of ECs are perturbed in *Yap/Taz* mutant vessels. Additionally, the *Yap/Taz* iEC-KO vasculature displayed aberrant vessel crossings (*Figure 4C,C',C'',D,D',D''*), suggesting that vessels may frequently have failed to anastomose or stabilize connections following sprouting, and instead passed each other. Interestingly, defects in cellular rearrangements, sprouting elongation and anastomosis have previously been associated with altered stability or dynamics of endothelial cell junctions (*Sauteur et al., 2014*; *Bentley et al., 2014*; *Giannotta et al., 2013*; *Lenard et al., 2013*; *Dejana et al., 2008*). The defects in vessel morphology were coupled to defects in function as *Yap/Taz* iEC-KO retinas displayed large haemorrhages from sprouts at the angiogenic front (*Figure 4E,E',F,F'*), indicating loss of junctional integrity.

Together, these results argue against the cell proliferation defect being the sole driver of the *Yap/Taz* iEC-KO phenotype and suggest that endothelial YAP/TAZ play a role in the regulation of EC junctions.

## YAP/TAZ regulate adherens junction morphology and stability

Staining for VE-Cadherin revealed several junctional alterations in *Yap/Taz* iEC-KO vessels (*Figure 4G–H*). In control retinas, cell junctions were thin and mostly linear (*Figure 4G'*), while in

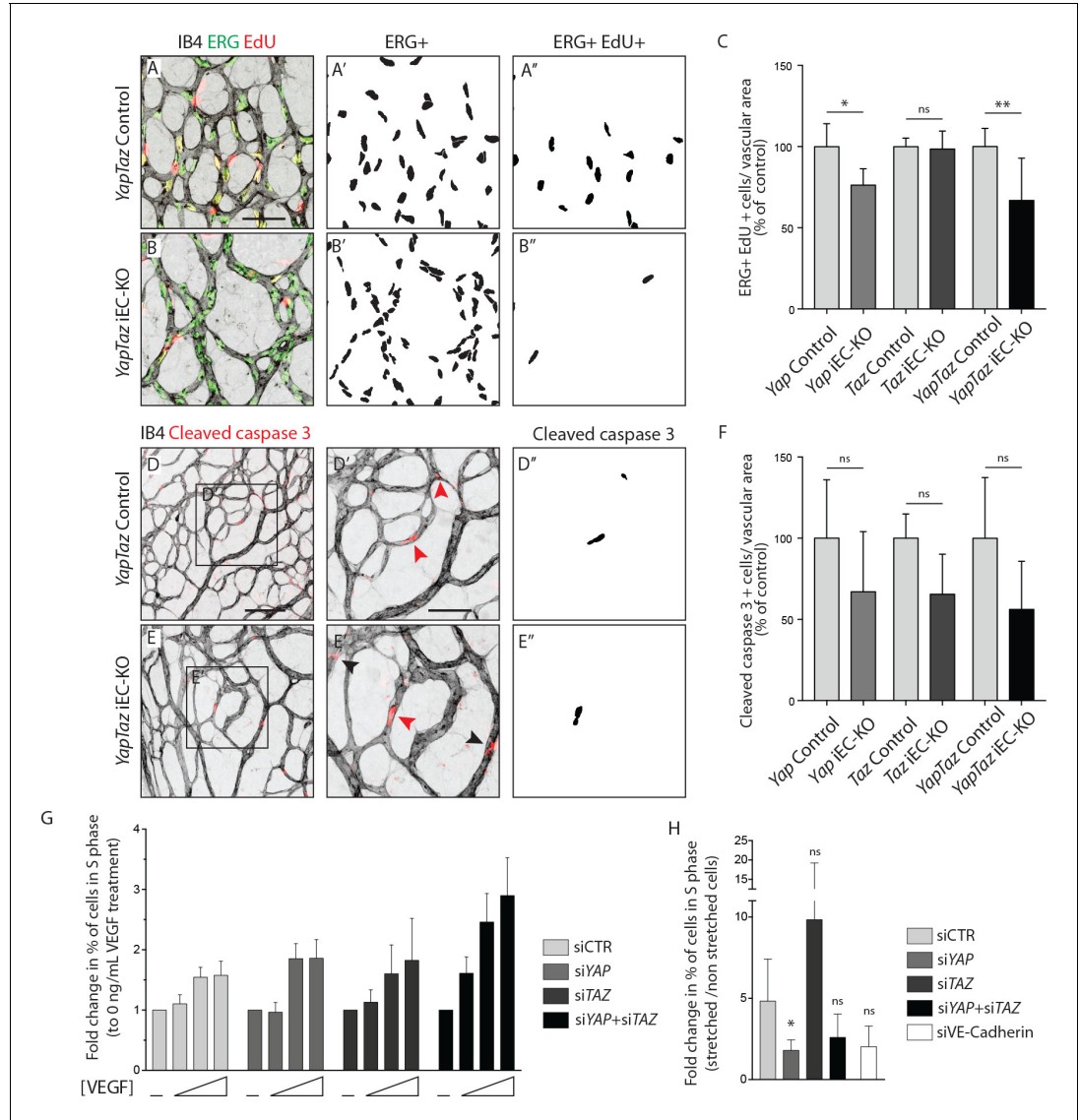

**Figure 3.** YAP and TAZ are required for endothelial cell proliferation in vivo and endothelial cell proliferation in response to mechanical stretch in vitro. (A, B) P6 retinal vessels labelled with IB4 (grey) and stained for EdU (red, marking S phase positive cells) and Erg (green, marking endothelial nuclei) in *YapTaz* iEC-KO (B) and littermate control mice (A). A',B', mask of Erg +cells indicating endothelial nuclei. (A'', B'') mask of Erg + and EdU + cells indicating proliferating endothelial cells. (C) Quantification of endothelial proliferation in *Yap* iEC-KO (n = 3 control/4 KO pups), *Taz* iEC-KO (n = 5 control/5 KO pups) and *YapTaz* iEC-KO (n = 8 control/7 KO pups). Number of EdU-positive and ERG-positive cells per IB4 labelled vascular area was calculated for each genotype and results are shown in percentage of the respective controls. Data are mean ±SD. p values were calculated using unpaired *t*-test. ns, p>0.05; *p<0.05; **p<0.01. Scale bar: 50 μm. (D, E) P6 retinal vessels labelled with IB4 (grey) and stained for cleaved caspase 3 (red) in *YapTaz* iEC-KO (E) and littermate control mice (D).D', E', magnification of boxed area in D,E. Red arrowheads, cleaved caspase 3 positive endothelial cell. Black arrowheads, cleaved caspase 3 outside vessels. D'',E'', mask of cleaved caspase 3 positive endothelial cells. (F) quantification of endothelial apoptosis in *Yap* iEC-KO (n = 7 control/7 KO pups), *Taz* iEC-KO (n = 4 control/4KO pups) and *YapTaz* iEC-KO (n = 5 control/4 KO pups). Data are mean ±SD. p values were calculated using unpaired *t*-test. ns, p>0.05. Scale bar: D-E 100 μm, D'-E' 50 μm. (G) Quantification of endothelial proliferation with increasing concentrations of VEGF treatment in YAP, TAZ and YAP/TAZ knockdown cells and control. HUVECs were treated with 0, 40, 200 or 1000 ng/mL VEGF for 24 hr and the percentage of cells in S phase was determined by flow cytometry. Graph shows the mean +SD fold change in percentage of S phase positive cells relative to 0 ng/mL of VEGF treatment. n = 3 independent experiments;>50.000 cells analysed per experiment per condition. (H) Quantification of endothelial proliferation after stretch in in YAP, TAZ, YAP/TAZ and VE-Cadherin knockdown cells and control. HUVECs were subjected to cyclic stretch for 24 hr and percentage of cells in S phase was determined by EdU pulsing and immunofluorescence staining. Graph shows the mean +SD fold change in percentage of S phase positive cells of stretched to non stretched cells for each knockdown condition. n = 5 independent experiments, >100 cells counted per experiment per condition. p values were calculated using unpaired *t*-test. ns, p>0.05; *p<0.05.
DOI: https://doi.org/10.7554/eLife.31037.008

*Figure 3 continued on next page*

*Figure 3 continued*

The following source data and figure supplements are available for figure 3:

**Source data 1.** Values for quantification of endothelial proliferation (*Figure 3C*) and apoptosis (*Figure 3F*) in P6 *Yap* iEC-KO, *Taz* iEC-KO and *YapTaz* iEC-KO and respective control pups.
DOI: https://doi.org/10.7554/eLife.31037.011
**Figure supplement 1.** YAP and TAZ proteins are lost after gene knockdown by siRNA in HUVECs.
DOI: https://doi.org/10.7554/eLife.31037.009
**Figure supplement 2.** VEGF treatment does not affect YAP and TAZ subcellular localisation.
DOI: https://doi.org/10.7554/eLife.31037.010

*Yap/Taz* iEC-KO retinas ECs displayed tortuous junctions (*Figure 4H'*). VE-Cadherin staining also unveiled profound differences in the arrangement of ECs within vessels. In control retinas, ECs were

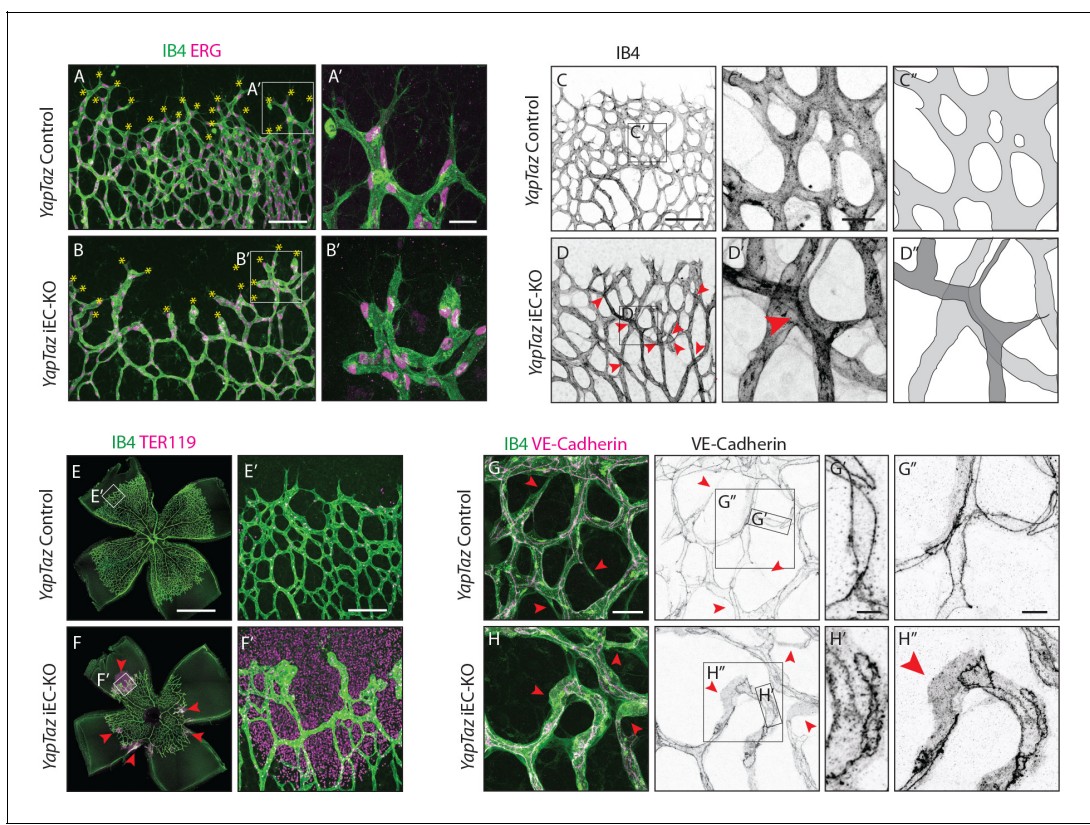

**Figure 4.** Combined loss of YAP and TAZ leads to decreased sprouting numbers and shape defects, vessel crosses, haemorrhages at the sprouting front and adherens junctions' defects in vivo. (**A, B**) P6 retinal vessels labelled with IB4 (green) and stained for ERG (magenta, marking endothelial nuclei) in *YapTaz* iEC-KO (**B**) and littermate control mice (**A**). Yellow asterisks mark sprouts. A',B', magnification of boxed areas in A and B. n = 9 control/9 KO pups. Scale bar: A,B 100 µm, A', B' 25 µm. (**C, D**) P6 retinal vessels labelled with IB4 in *YapTaz* iEC-KO (**D**) and littermate control mice (**E**). Red arrowheads, vessel crosses. (**C', D'**) magnification of boxed areas in C,D. C'',D'', depiction of vessels in C' and D'; different colours represent vessels in different 3D planes. n = 4 control/4 KO pups. Scale bar: C,D 100 µm, C'-D' 20 µm. (**E, F**) P6 retinal vessels labelled with IB4 (green) and stained for TER119 (magenta, marking red blood cells) in *YapTaz* iEC-KO (**F**) and littermate control mice (**E**). Red arrowheads, haemorrhages. E',F', magnification of boxed areas in E and F. n = 4 control/5 KO pups. Scale bar: E,F 1000 µm, E', F' 100 µm. (**G, H**), P6 retinal vessels labelled with IB4 (green) and stained for VE-Cadherin (magenta) in *YapTaz* iEC-KO (**H**) and littermate control mice (**G**). Red arrowheads, no longitudinal VE-Cadherin labelled junction along vessel axis denoting unicellular vessel segments. (**G',H', G'',H''**) magnification of boxed areas in G and H. n = 4 control/4 KO pups. Scale bar: G,H 25 µm, G',H' 5 µm, G'',H'' 10 µm.
DOI: https://doi.org/10.7554/eLife.31037.012

The following figure supplement is available for figure 4:

**Figure supplement 1.** Combined loss of YAP and TAZ leads to decreased number of sprouts in the developing mouse retina.
DOI: https://doi.org/10.7554/eLife.31037.013

arranged into multicellular tubes, highlighted by the presence of two or more VE-Cadherin junctions running longitudinally along the axis of the vessels (*Figure 4G,G''*). Some unicellular segments lacking VE-Cadherin staining could also be found and always correlated with decreasing calibre, indicative of regressing vessels (*Figure 4G* red arrowheads) (*Franco et al., 2015*). In contrast, in *Yap/Taz* iEC-KO retinas we observed many unicellular vessel segments lacking longitudinal VE-Cadherin junctions, but in vessels of normal calibre (*Figure 4H*, red arrowheads and H''). As junctional remodelling has been shown to be required for the cellular rearrangements that establish multicellular tubes (*Sauteur et al., 2014*), these results suggest that YAP and TAZ regulate junctional remodelling.

VE-Cadherin staining in HUVECs after YAP, TAZ and YAP/TAZ knockdown further revealed altered junctional morphologies compared to control cells. Previous studies have correlated junctional morphology with cellular activities. In vivo, straight or linear junctions were associated with high Notch activity and stalk cell behaviour, while serrated junctions (also referred to as VE-Cadherin fingers) were found in tip cells or actively rearranging cells (*Bentley et al., 2014*). In vitro, VE-Cadherin fingers were shown to steer migrating ECs and couple leader and follower cells (*Hayer et al., 2016*), and have also been correlated with increased permeability in cell monolayers. More recently, junction associated intermediate lamellipodia (JAIL) have been identified in the sprouting vessels of the mouse retina, and linked to increased migration (*Cao et al., 2017*) as well as decreased permeability in cultured ECs (*Breslin et al., 2015*).

To more accurately describe the differences in junctional morphology after YAP/TAZ knockdown, we defined five junctional categories: straight junctions, thick junctions, thick to reticular junctions, reticular junctions and fingers (*Figure 5E*). Live imaging analysis of VE-Cadherin-GFP transduced HUVECs showed that reticular junctions correspond to JAIL, and thick to reticular junctions to small JAIL. (*Figure 5—video 1*). Whereas control cells showed mostly reticular junctions (*Figure 5A,F*), the knockdown of YAP and TAZ led to an increase in straight junctions and fingers, respectively (*Figure 5B,C,F*). The combined knockdown of YAP/TAZ led to an increase in both straight junctions and fingers and to a loss of reticular junctions (*Figure 5D,F*). In addition, the knockdown of YAP/TAZ led to junctional breaks in the monolayer, as seen by the presence of gaps in VE-Cadherin stainings (*Figure 5D*, red arrowheads). Together, these observations demonstrate that YAP and TAZ together are required for the formation of JAIL and reduce the formation of straight junctions and fingers.

To understand whether this shift in morphology translated into a functional defect, we investigated the permeability of the monolayer to 250 kDa dextran molecules. Only the combined knockdown of YAP/TAZ led to a significant increase in permeability in comparison to the control situation (*Figure 5G*), suggesting that YAP/TAZ are both required for the barrier function of the endothelium and can compensate for each other in this particular role.

The dynamic rearrangements of ECs during sprouting require that cell-cell junctions are constantly assembled, rearranged and disassembled. To understand whether YAP and TAZ regulate the turnover of cell junctions, we pulse-labeled VE-Cadherin molecules at cell junctions using an antibody directly coupled to a fluorescent dye for 30 min (*Figure 5H–I*) (*Dorland et al., 2016*). The antibody was subsequently washed out and cells cultured for two more hours in normal conditions, before being fixed and stained for surface VE-Cadherin using a second fluorescent label. Comparing the two sequential VE-cadherin labels allowed us to distinguish junctions with high, intermediate and low turnover rates (*Figure 5J*). In control cells, 44% of patches were of high turnover junctions, 24% of intermediate turnover junctions and 32% of low turnover junctions (*Figure 5K*). The knockdown of YAP/TAZ significantly decreased the percentage of high turnover junctions to 14% (p=0.0387) and increased the percentage of low turnover junctions to 58%. Interestingly, we found a correlation between the morphology of junctions and VE-Cadherin turnover rates (*Figure 5L*): straight junctions and fingers showed the lowest turnover rate, while reticular junctions showed the highest. To understand if the different VE-Cadherin turnover observed after knockdown of YAP/TAZ was caused by a shift in morphology, we compared the turnover of VE-Cadherin within the same morphological categories. Knockdown of YAP/TAZ decreased the percentage of high turnover junctions within all morphological categories, confirming a specific defect in VE-Cadherin turnover. To further investigate how YAP and TAZ affect the turnover of VE-cadherin, we transfected HUVECs with a photo-convertible fluorescent protein tagged VE-Cadherin (VE-cadherin-mEos3.2) and measured the fluorescence loss after photo conversion in straight junctions. The knockdown of YAP/TAZ led to a significant increase in the amount of immobile VE-cadherin-mEos3.2 at the junctions (*Figure 5M*) without significantly affecting the half time of fluorescence loss of the mobile fraction (*Figure 5N*). Together,

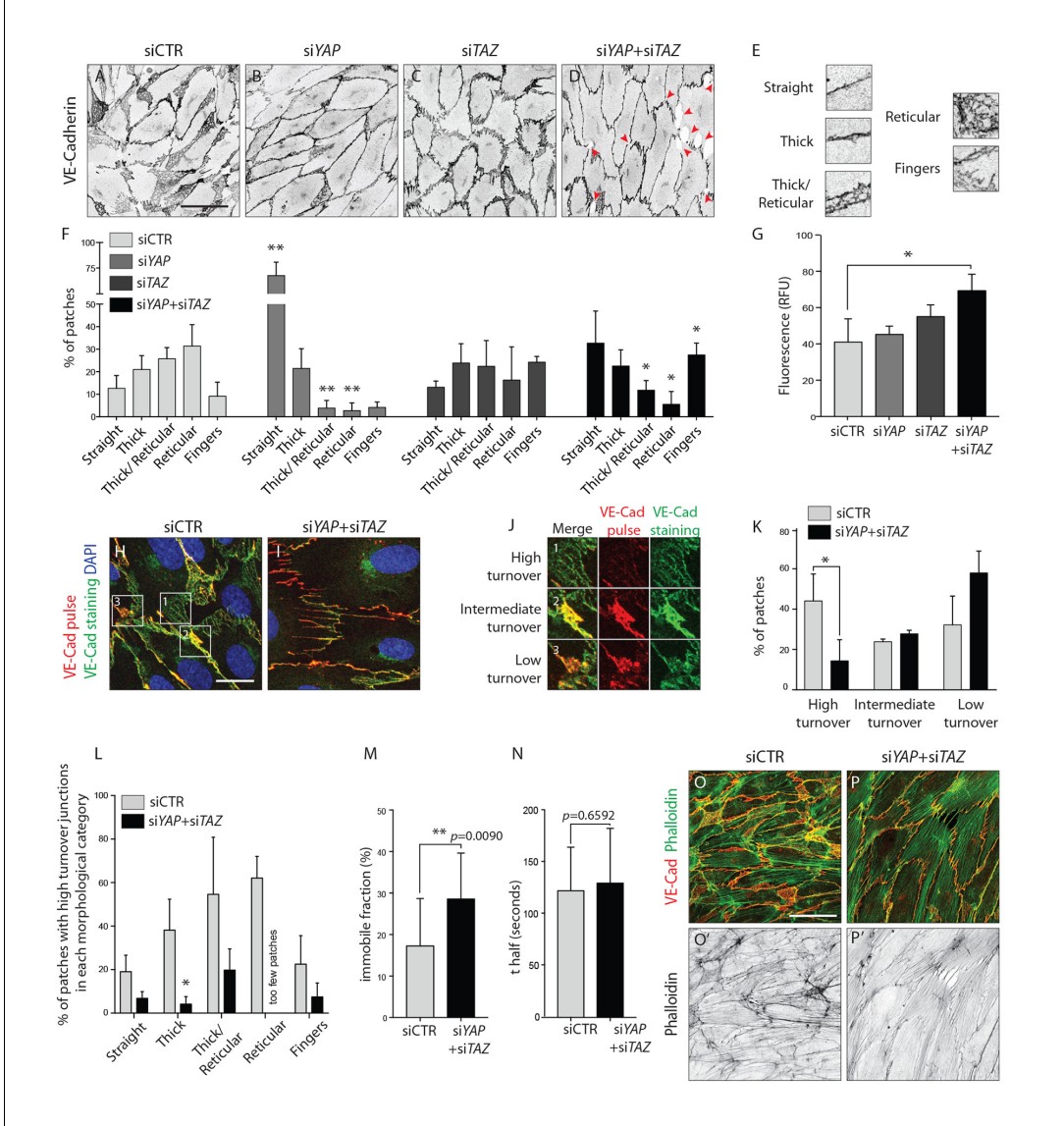

**Figure 5.** YAP and TAZ regulate adherens junctions' morphology, monolayer permeability and VE-Cadherin turnover in vitro. (A–D) HUVECs knocked down for YAP (B), TAZ (C) and YAP/TAZ (D) and control (A) stained for VE-Cadherin. Red arrowheads, discontinuous VE-Cadherin. Scale bar: 50 μm. (E) Representative patches used for manual morphological classification of adherens junctions in five categories: straight junctions, thick junctions, thick to reticular junctions, reticular junctions and fingers. (F) Morphological analysis of VE-Cadherin labelled cell junctions in HUVECs knocked down for YAP, TAZ and YAP/TAZ. Data are mean percentage ±SD of 3 independent experiments (two for siTAZ). n > 140 patches of VE-Cadherin stained HUVECs per knockdown condition per experiment. p values were calculated using unpaired t-test between knocked down cells for YAP, TAZ and YAP/TAZ and control. *p<0.05; **p<0.01. (G) Permeability of YAP, TAZ and YAP/TAZ knockdown monolayers of HUVECs to 250 kDa fluorescent dextran molecules. Data are mean +SD of 3 independent experiments. p values were calculated using unpaired t-test between knocked down cells for YAP, TAZ and YAP/TAZ and control. RFU, relative fluorescence units. *p<0.05. (H, I) HUVECs knocked down for YAP/TAZ (I) and control (H) triple labelled with DAPI (blue), pulsed VE-Cadherin 55-7HI (red, VE-Cadherin pulse), and surface VE-Cadherin (green, VE-Cadherin staining). VE-Cadherin 55-7HI pulse was done for 30 min and cells were fixed 2 hr after end of pulse. Scale bar: 20 μm. (J) Representative patches used for manual classification of junctions into high, intermediate and low turnover. (K) Quantification of junctional turnover in YAP/TAZ knockdown cells and control. (L) Quantification of the percentage of high turnover junctions in each morphological category in YAP/TAZ knockdown cells and control. (K, L) Data are mean ±SD of 3 independent experiments. n > 70 patches per knockdown condition per experiment. Fewer then five patches were reticular in YAP/TAZ knockdown, not allowing for reliable assessment of percentages between high, intermediate and low turnover. p values were calculated using unpaired t-test. *p<0.05. (M, N) Fluorescence loss after photoconversion of VE-Cadherin mEos in straight junctions of YAP/TAZ knockdown HUVECs and control HUVECs. M, VE-Cadherin mEos immobile fraction. N, VE-Cadherin mEos half-life of fluorescence loss. Data are mean ±SD of 3 independent experiments. n = 15 control cells and 16 cells YAP/TAZ knockdown cells. p values were calculated using unpaired t-test between knocked down cells for YAP/TAZ and control.

*Figure 5 continued on next page*

*Figure 5 continued*

**p<0.01. (**O**, **P**) HUVECs knocked down for YAP/TAZ (**P**) and control (**O**) double stained for VE-Cadherin (red) and f-actin (green, phalloidin). (**O'**, **P'**) f-actin (black, phalloidin). Scale bar: 50 μm.

DOI: https://doi.org/10.7554/eLife.31037.014

The following video and source data are available for figure 5:

**Source data 1.** Values for quantification of morphological (*Figure 5F*) and junctional turnover (*Figure 5K*) analysis of VE-Cadherin in HUVECs knocked down for YAP, TAZ and YAP/TAZ.

DOI: https://doi.org/10.7554/eLife.31037.015

**Figure 5—video 1.** Reticular junctions correspond to junction associated intermediate lamellipodia.

DOI: https://doi.org/10.7554/eLife.31037.016

these results show that YAP/TAZ increase the turnover of adherens junctions by increasing the pool of mobile VE-Cadherin.

Immunofluorescence staining for f-actin in YAP/TAZ deficient cells (*Figure 5P,P'*) showed an increase in actin bundles (actin filaments along the junction and stress fibers) and a decrease in branched actin networks, in comparison with control cells (*Figure 5O,O'*). This result correlates with the junctional morphologies observed before, as straight junctions and fingers associate respectively with actin filaments and stress fibers (*Abu Taha and Schnittler, 2014*), while JAILs associate with branched actin (*Abu Taha et al., 2014*). Interestingly, our VE-Cadherin pulse labeling experiments showed that the junction morphologies with less turnover, straight junctions and fingers, were the ones associated with bundled actin filaments, while the morphologies with higher turnover, reticular junctions, associated with branched actin networks.

As cell junctions and the actin cytoskeleton are essential for ECs to rearrange and migrate collectively (*Vitorino and Meyer, 2008*), and *Yap/Taz* iEC-KO retinas presented less elongated sprouts suggestive of a migration defect, we asked whether cell migration was also regulated by YAP/TAZ.

## YAP/TAZ are required for individual endothelial cell migration

To address the requirement of YAP and TAZ for endothelial cell migration, we performed scratch-wound assays (*Figure 6A–H*). While in the control situation the wound was completely closed at 16 hr (*Figure 6A,B,I*), less than 50% of the wound area was closed after YAP knockdown at the same time point (p<0.0001) (*Figure 6C,D,I*). A stronger effect on endothelial cell migration was observed after the knockdown of TAZ and YAP/TAZ, with less than 20% of the wound area being closed (p<0.0001 for siTAZ vs siCTR and for siYAP +siTAZ vs siCTR) (*Figure 6E,F,G,H,I*).

Given that cells aggregated at the sprouting front of *Yap/Taz* iEC-KO retinas, we wondered whether in addition to defective directional cell migration they also lacked the ability to shuffle with the neighbouring cells. Recent data illustrated that collectively migrating ECs in vitro move in streams and swirls and display straight junctions along the lateral boundaries and fingers along the front and rear (*Hayer et al., 2016*). To investigate collective cell migration we therefore analysed the arrangement of cells in a confluent monolayer (*Figure 6J–O*). Control cells displayed a cobble-stone appearance without identifiable subgroups of cells (*Figure 6J*). In contrast, after knockdown of YAP/TAZ cells adopted elongated shapes and arranged into streams and swirls (*Figure 6K*). To quantity this effect we used the longest axis of the EC nucleus as a proxy for the orientation of each cell and developed a measure of monolayer coordination based on the alignment of cells with their neighbours (*Figure 6L,L',M,M', N*). A score of 1 would signify parallel alignment between all cells, and a score of 0 random alignment of the population. Control cells displayed higher than random alignment with their closest neighbours, but cells beyond 300 μm from each other were arranged at random (*Figure 6I*). While the knockdown of YAP did not affect the alignment score of cells, the knockdown of TAZ led to increased alignment. The combined knockdown of YAP/TAZ led to an even higher degree of coordination, with higher alignment scores across all distances between cells. These results suggest that YAP/TAZ promote the ability of cells to distribute individually within monolayers.

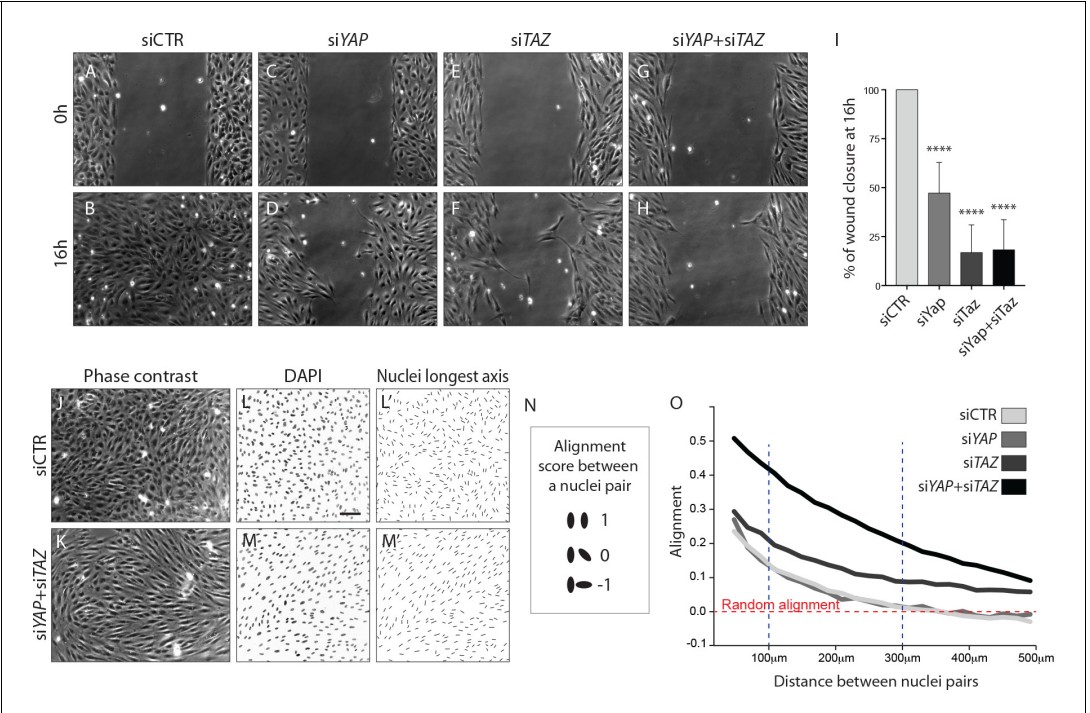

**Figure 6.** YAP and TAZ are required for uncoupled, individual cell migration. (A–H), Phase contrast images of YAP (C,D), TAZ (E,F) and YAP/TAZ (G,H) knockdown HUVECs and control (A,B) immediately after removing barrier to create a cell free space (A,C,E,G) and 16 hr later (B,D,F,H). (I), Quantification of wound closure at 16 hr. Data are mean ±SD of 3 independent experiments (8–9 biological replicates). *p* values were calculated using unpaired *t*-test between knocked down cells for YAP, TAZ or YAP/TAZ and control. ****p<0.0001. (J, K), Phase contrast images of YAP/TAZ knockdown monolayer of HUVECs (K) and control (J). (L, M), Fluorescence labelling of nuclei with DAPI of YAP/TAZ knockdown monolayer of HUVECs (M) and control (L). Scale bar: 100 mm. (L',M') Longest axis of nuclei. (N) Alignment score between nuclei pairs used for quantification of cell coordination in O. Angles made by the nuclei longest axis of a pair of nuclei were calculated; angles of 0, 45 and 90 degrees scored 1,0 and −1 in alignment. (O) Coordination plot of monolayers of HUVECs knocked down for YAP, TAZ and YAP and TAZ and control. Graph shows mean alignment score of all pairs of cells in the monolayer plotted against distance between them. Randomly aligned cells score 0 in mean alignment. n = 3 independent experiments, >10.000 pairs of nuclei analysed per knockdown condition per experiment.

DOI: https://doi.org/10.7554/eLife.31037.017

The following source data is available for figure 6:

**Source data 1.** Values for quantification of wound closure at 16 hr in YAP, TAZ and YAP/TAZ knockdown HUVECs and control (*Figure 6I*).

DOI: https://doi.org/10.7554/eLife.31037.018

## Nuclear YAP and TAZ inhibit Notch and BMP signalling in endothelial cells

To gain insight into the nuclear function of YAP and TAZ, we generated a *Pdgfb-iCreERT2* -inducible TAZ gain-of-function mouse allele, in which a mutated version of TAZ (TAZ S89A) is introduced in the *Rosa26* locus and expressed by a *CAG* promoter following Cre-mediated excision of an upstream stop cassette (*Figure 7—figure supplement 1*). The serine-to-alanine mutation in TAZ results in enhanced nuclear TAZ localization as it escapes phosphorylation by the upstream Hippo kinase cascade (*Varelas et al., 2010*). The allele also expresses nuclear EGFP by means of an IRES sequence, allowing the identification of recombined cells expressing the TAZ mutant protein. *Taz* iEC-GOF retinas exhibited 25% increased sprouting (±12.2, p=0.0074) (*Figure 7A,B* yellow asterisks and *Figure 7C*) and 19% increased branching (±8.3, p=0.0012) (*Figure 7D*). Thus, driving nuclear TAZ expression leads in many aspects to the opposite phenotype of *Yap/Taz* iEC-KO retinas, suggesting that the loss of the nuclear function of YAP/TAZ plays a key role in the development of the observed vascular loss-of-function phenotypes.

To elucidate the transcriptional targets of YAP and TAZ, we performed unbiased transcriptome analysis on HUVECs transduced with adenoviruses encoding for *YAP* and *TAZ* gain-of-function mutants or *GFP* as a control (AdYAP$^{S127A}$, AdTAZ$^{S89A}$, AdGFP) (*Figure 7—figure supplement 2*).

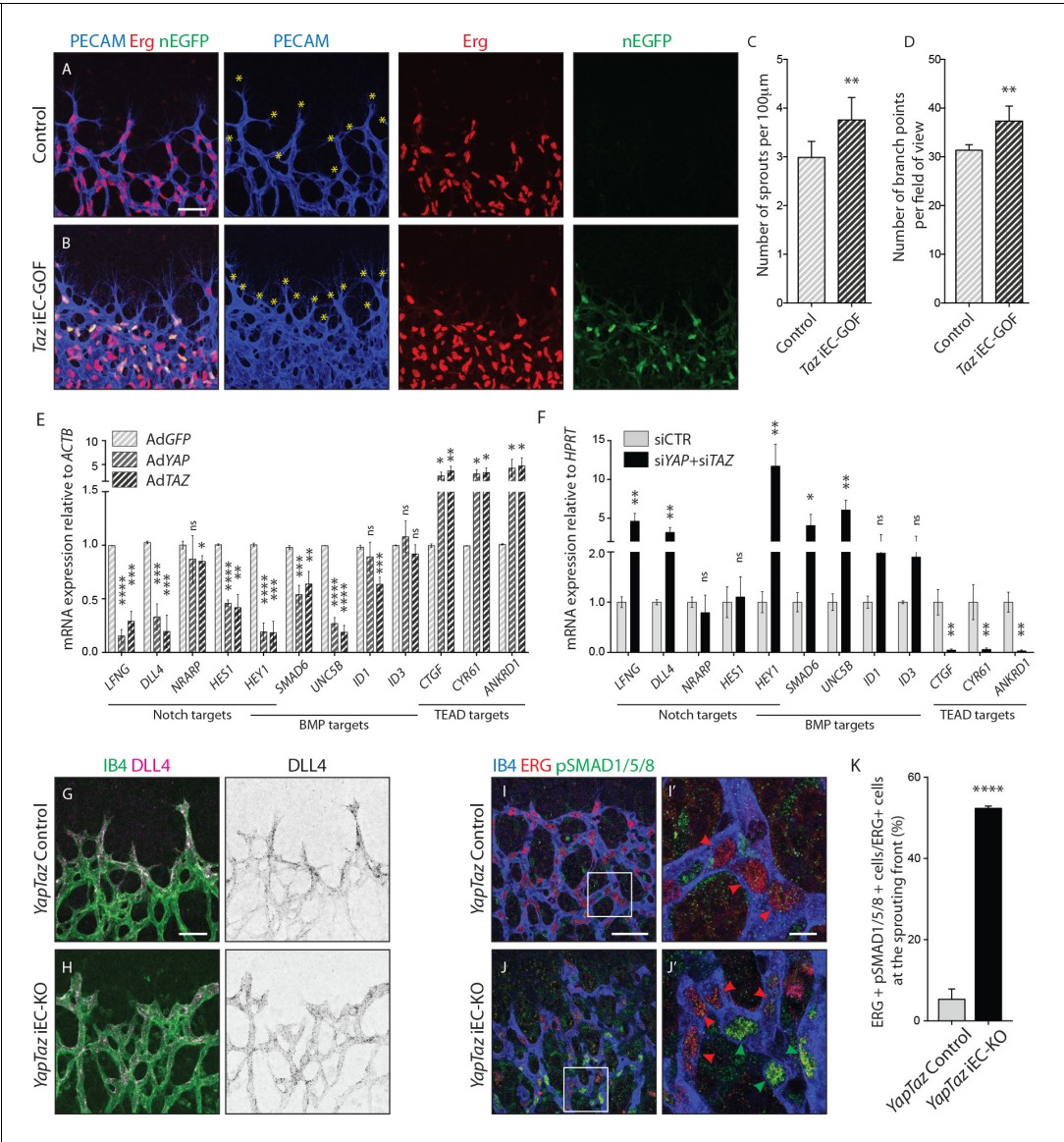

**Figure 7.** Nuclear YAP and TAZ inhibit Notch and BMP signalling in endothelial cells. (**A–B**) Retinas from P6 *Taz* iEC-GOF (**B**) and control pups (**A**) were stained for the endothelial marker PECAM (blue) and the endothelial nuclei marker ERG (red). *Taz* iEC-GOF mice express mosaically nuclear *EGFP* (nEGFP, green) marking cells expressing the TAZ gain of function mutation TAZS89A. Yellow asterisks mark sprouts. Images correspond to maximum projection of z stack. Scale bar: 50 µm. (**C**) Quantification of number of sprouts per 100 µm of sprouting front extension at P6 in *Taz* iEC-GOF mice (n = 6 pups) and littermate control mice (n = 6 pups). Data are mean ±SD. *p* values were calculated using unpaired *t*-test. **p<0.01. (**D**) Quantification of branching frequency (i.e. number of branching points per field of view) in *Taz* iEC-GOF mice (n = 6 pups) and littermate control mice (n = 6 pups). Data are mean ±SD. *p* values were calculated using unpaired *t*-test. **p<0.01. (**E**), Reverse transcriptase PCR of HUVECs transduced with adenoviruses carrying YAP (Ad*YAP*) and TAZ (Ad*TAZ*) constitutively active forms and control (Ad*GFP*). Data are mean ±SD of 3 independent experiments. *p* values were calculated using unpaired *t*-test between Ad*GFP* and Ad*YAP* or Ad*TAZ*. *p<0.05; **p<0.01; ***p<0.001; ****p<0.0001. (**F**) Reverse transcriptase PCR of YAP/TAZ knockdown HUVECs and control. Data are mean ±SD of 3 independent experiments. *p* values were calculated using unpaired *t*-test. *p<0.05; **p<0.01; ***p<0.001; ****p<0.0001. (**G,H**), P6 retinal vessels labelled with IB4 (green) and stained for DLL4 (magenta) in *YapTaz* iEC-KO mice (**H**) and littermate control mice (**G**). Images correspond to maximum projection of z stack. Scale bar: 50 µm. (**I,J**), P6 retinal vessels labelled with IB4 (blue) and stained for ERG (red, marking endothelial nuclei) and pSMAD1/5/8 (green) in *YapTaz* iEC-KO (**J**) and littermate control mice (**I**). Images correspond to single confocal planes. (**I',J'**) magnification of boxed areas in I and J. Red arrowheads, endothelial nuclei negative for pSMAD1/5/8. Green arrowheads, endothelial nuclei positive for pSMAD1/5/8. Scale bar: I,J 50 µm, I', J' 10 µm. (**K**) Quantification of endothelial cells positive for pSMAD1/5/8 at the sprouting front of the P6 retina in *YapTaz* iEC-KO (n = 3 pups) and littermate control mice (n = 3 pups). Data are mean percentage ±SD. *p* values were calculated using unpaired *t*-test. ****p<0.0001.

DOI: https://doi.org/10.7554/eLife.31037.019

The following source data and figure supplements are available for figure 7:

*Figure 7 continued on next page*

*Figure 7 continued*

**Source data 1.** Values for quantification of number of sprouts (*Figure 7C*) and branching frequency (*Figure 7D*) in *Taz* iEC-GOF mice and controls.
DOI: https://doi.org/10.7554/eLife.31037.024
**Figure supplement 1.** Targeting strategy used for the generation of the conditional TAZ gain-of-function mouse model.
DOI: https://doi.org/10.7554/eLife.31037.020
**Figure supplement 2.** Microarray of YAP and TAZ gain of function mutant cells.
DOI: https://doi.org/10.7554/eLife.31037.021
**Figure supplement 3.** YAP and TAZ knockdown increases Notch and BMP reporter activities in vitro.
DOI: https://doi.org/10.7554/eLife.31037.022
**Figure supplement 4.** DLL4 intensity in *YapTaz* iEC-KO.
DOI: https://doi.org/10.7554/eLife.31037.023

Forced activation of YAP and TAZ led to congruent gene expression changes including the canonical YAP/TAZ target genes *CYR61*, *ANKRD1*, and *CTGF*, as expected. Interestingly, YAP and TAZ also suppressed numerous Notch and BMP target genes. During sprouting angiogenesis, Notch and BMP9/10 signalling restrict the acquisition of a tip cell phenotype by activated ECs (*Hellström et al., 2007*; *Lobov et al., 2007*; *Suchting et al., 2007*; *Siekmann and Lawson, 2007*; *David et al., 2008*; *Larrivée et al., 2012*; *Laux et al., 2013*). These results were confirmed by qRT-PCR analysis (*Figure 7E*): AdYAP^S127A and AdTAZ^S89A cells expressed significantly less *LFNG*, *DLL4* and *HES1* (Notch target genes), *SMAD6*, *UNC5B* and *ID1* (BMP target genes) and *HEY1* (a common Notch and BMP target gene) than control cells. Consistent with these findings, knockdown of YAP or TAZ lead to a substantial increase in Notch reporter activity (*Figure 7—figure supplement 3A*) and target gene expression (*Figure 7F*). Similar effects were observed for the BMP pathway (*Figure 7—figure supplement 3B* and *Figure 7F*), while TEAD-driven reporter activity and YAP/TAZ target genes were repressed (*Figure 7—figure supplement 3C* and *Figure 7F*).

To understand if Notch and BMP signalling were also affected in vivo, we stained *Yap/Taz* iEC-KO retinas for DLL4 and phospho-SMAD1/5/8. In control retinas, DLL4 expression was highest at the leading edge, decreasing over the first 100 µm from the sprouting front, beyond which the expression was evenly low throughout the vessels in the plexus (*Figure 7G,H* and *Figure 7—figure supplement 4*). In *Yap/Taz* iEC-KO retinas the expression of DLL4 at the sprouting front was higher; additionally, the area of high DLL4 was broader, decreasing for up to 200 µm from the sprouting front before flattening to the lower levels of the plexus. Moreover, staining *Yap/Taz* iEC-KO retinas for pSMAD1/5/8 showed a ~ 10 fold increase in the number of ECs positive for pSMAD1/5/8 at the sprouting front (p<0.0001) (*Figure 7I,I', J,J',K*).

Together, these results identify that endothelial YAP and TAZ repress Notch and BMP signalling during angiogenesis and retinal vascular expansion.

## Inhibition of BMP signalling partially rescues the cellular defects of YAP/TAZ loss in endothelial cells

To address whether the Notch or BMP signalling increase were responsible for the cellular defects of YAP/TAZ deficiency, we used different inhibitors to manipulate Notch and BMP signalling in cultured cells. Treatment with the Notch inhibitor DBZ corrected the Notch reporter activity increase after YAP and TAZ knockdown in HUVECs (*Figure 8A*), but did not affect the migration of cells in the scratch wound assay (*Figure 8B*).

To decrease BMP signaling we screened several inhibitors targeting different BMPs or BMP receptors (*Figure 8C*, *Supplementary file 1*). Only Ldn193187, an Alk2/3/1 kinase inhibitor, significantly decreased the BMP reporter activity after YAP and TAZ knockdown (p=0.0063) (*Figure 8D*). In comparison to control cells, the BMP reporter activity in YAP/TAZ deficient cells was 12 times higher, dropping to seven times higher after Ldn193187 treatment. To understand if the partial BMP rescue altered the cellular defects after YAP/TAZ loss, we performed the scratch wound and permeability assays using this inhibitor. Ldn193187 treatment partially rescued the migration defect of YAP/TAZ knockdown HUVECs in the scratch wound assay, significantly increasing the wound closure at 16 hr from 33 ± 3% in DMSO treated cells to 56 ± 5% in Ldn193187 treated cells (p=0.0022) (*Figure 8E*). Ldn193187 treatment fully rescued the permeability defect of YAP/TAZ knockdown HUVECs to 250 kDa FITC dextran molecules (*Figure 8F*). Staining for VE-Cadherin further showed

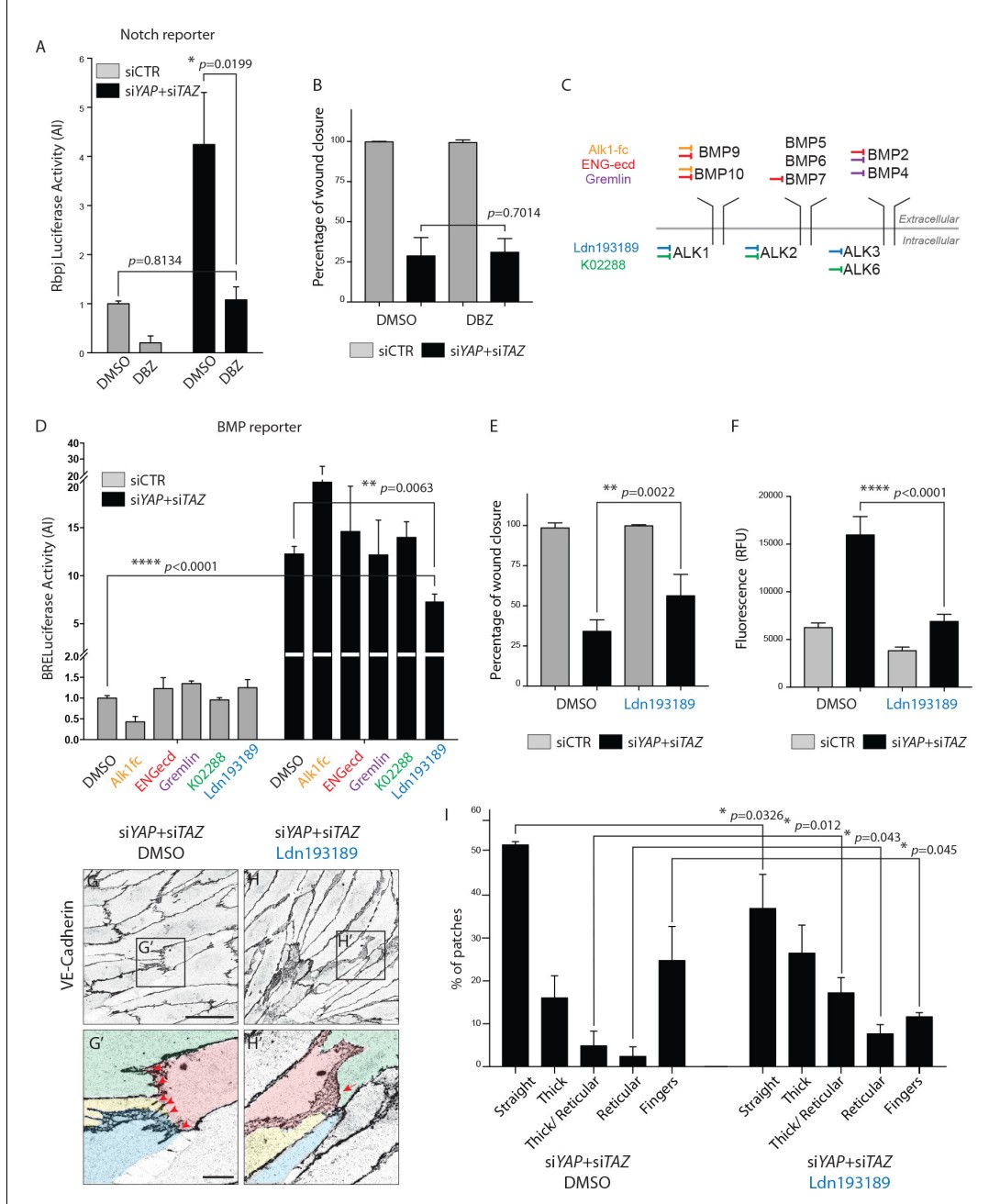

**Figure 8.** BMP inhibition partially rescues the cellular defects of YAPTAZ knockdown HUVECs. (**A**) Luciferase reporter assay for Notch activity in YAP/TAZ knockdown HUVECs and controls treated with 0.1 μM DBZ or DMSO. Data are mean ±SEM. *p* values were calculated using unpaired *t*-test. n ≥ 3 biological replicates. (**B**) Quantification of wound closure at 16 hr for HUVECs knocked down for YAP/TAZ and treated with 0.1 μM DBZ or DMSO. Data are mean ±SD. *p* values were calculated using unpaired *t*-test. n = 6 biological replicates. (**C**) Schematic of the BMP inhibitors used depicting preferential sites of inhibition. Alk1fc, ENGecd and Gremlin preferentially bind extracellular BMPs. K02288 and Ldn193189 are kinase inhibitors. (**D**) Luciferase reporter assay for BMP activity in YAP/TAZ knockdown HUVECs and controls treated with 25 ng/mL Alk1fc, 0.25 μg/mL ENGecd, 0.1 μg/mL Gremlin, 1 μM K02288, 1 μM Ldn193189 and DMSO. Data are mean ±SEM. *p* values were calculated using unpaired *t*-test. n ≥ 3 biological replicates. (**E**) Quantification of wound closure at 16 hr for HUVECs knocked down for YAP/TAZ and treated with 1 μM Ldn193189 or DMSO. Data are mean ±SD. *p* values were calculated using unpaired *t*-test. n = 6–7 biological replicates. (**F**) Permeability of HUVECs knocked down for YAP/TAZ and treated with 1 μM Ldn193189 or DMSO to 250 kDa fluorescent dextran molecules. Data are mean ±SD. RFU, relative fluorescence units. *p* values were calculated using unpaired *t*-test. n = 6 biological replicates. (**G, H**) HUVECs knocked down for YAP/TAZ and treated with 1 μM Ldn193189 (H) or DMSO (G) stained for VE-Cadherin. (**G', H'**) different colours mark different cells. Red arrowheads, fingers. Red arrow, reticular junction. Scale bar G, H, 50 μm. Scale bar G', H', 10 μm. (**I**) Morphological analysis of VE-Cadherin labelled cell junctions in HUVECs knocked down for YAP/TAZ and treated with 1 μM Ldn193189 or

*Figure 8 continued on next page*

*Figure 8 continued*

DMSO control. Data are mean ±SD. p values were calculated using unpaired t-test. n = 3 biological replicates; n ≥ 45 patches of VE-Cadherin stained HUVECs per condition per replicate.

DOI: https://doi.org/10.7554/eLife.31037.025

The following source data and figure supplement are available for figure 8:

**Source data 1.** Values of luciferase reporter assays for Notch (*Figure 8A*) and BMP (*Figure 8D*) activity in YAP/TAZ knockdown HUVECs and controls treated with Notch or BMP inhibitors.

DOI: https://doi.org/10.7554/eLife.31037.027

**Figure supplement 1.** Nuclear YAP and TAZ increase the expression of BMP inhibitors.

DOI: https://doi.org/10.7554/eLife.31037.026

that Ldn193187 treatment decreased the frequency of finger junctions and increased the frequency of JAIL in YAP/TAZ deficient cells (*Figure 8G–H*). Quantification of junction morphologies identified that Ldn193187 treatment decreased the frequency of finger junctions in YAP/TAZ deficient cells to the levels found in control cells (*Figure 8I* and *Figure 5F*, % of fingers: siYAP/TAZ DMSO, 25 ± 5%, siYAP/TAZ Ldn193187 12 ± 0.6%, siCTR: 9 ± 4%). Ldn193187 treatment also led to a 3-fold increase in thick to reticular (p=0.012) and reticular junctions (p=0.043) in YAP/TAZ knockdown cells. The frequency of straight junctions was also significantly decreased (p=0.0326) although still remaining high in comparison to control cells. Together, these results show that the BMP signaling increase is at least partially involved in the cellular defects caused by YAP/TAZ deficiency.

To gain insight into how YAP and TAZ repress BMP signalling, we analysed the gene expression of BMPs, BMP receptors and co-receptors and BMP antagonists in *YAP* and *TAZ* gain of function mutant HUVECs (*Figure 8—figure supplement 1*). As YAP and TAZ function as transcriptional co-activators, the direct targets of YAP and TAZ would be upregulated in this assay, while genes that are downregulated would represent indirect regulation. Forced activation of YAP and TAZ led to increased expression of the BMP ligands BMP2, BMP4 and BMP6, which cannot explain the increase in BMP signalling in the loss of function condition. However, we also found increased expression of the BMP antagonists *FST*, *CTGF*, *BAMBI*, *SMURF2*, *NOG* and *SMURF1*. Thus YAP and TAZ decrease BMP signalling in endothelial cells possibly by increasing the expression of BMP inhibitors.

Together, these results suggest that YAP/TAZ repress BMP activation in endothelial cells, modulating junctions and cell migration.

## Discussion

The present study aimed to provide a detailed understanding of the distribution and function of endothelial YAP and TAZ in angiogenesis. Our finding that YAP and TAZ were present in the nucleus of ECs at the sprouting front of developing vessels shows parallels with other cell types where nuclear YAP and TAZ are detected in actively proliferating areas of developing tissues. Interestingly, however, YAP and TAZ show distinct expression patterns in ECs, although these proteins show a high degree of redundancy in many other cell types. While TAZ was predominantly expressed in the sprouting front where it accumulated strongly in endothelial nuclei, YAP was mostly cytoplasmic both in the sprouting front and also in more mature, remodelling vessels. Interestingly, recent work by Sakabe and colleagues showed that cytoplasmic YAP promotes endothelial cell migration (*Sakabe et al., 2017*).

In addition to nuclear and cytoplasmic YAP and TAZ, we also detected junctional localization of these proteins in retinal vessels. A previous study by Giampietro and colleagues (*Giampietro et al., 2015*) has shown that endothelial YAP associates with adherens junction proteins at stable junctions and that this prevents its nuclear accumulation and transcriptional activity. Whether this is also true for TAZ has previously not been addressed. A sequestration of YAP and TAZ either in the cytoplasm or bound to junctional proteins can potentially serve different and not necessarily mutually exclusive roles: preventing their nuclear activity, keeping a pool of protein ready to shuttle to the nucleus and drive gene expression, and having other cytoplasmic functions. It is not yet entirely clear what regulates the subcellular localisation of YAP and TAZ in the developing vasculature. ECs at the sprouting front and in more mature vessels have different adherens junctions, experience distinct levels of signalling from secreted angiogenic molecules and are exposed to different levels of shear stress by

blood flow. In vitro, endothelial YAP and TAZ relocate to the nucleus upon disruption of cell junctions or loss of VE-Cadherin (shown for YAP by Choi and colleagues (*Choi et al., 2015*) and confirmed in our analysis also for TAZ, data not shown). In the mouse retina vasculature, Cao and colleagues (*Cao et al., 2017*) recently observed that endothelial cells at the sprouting front display reduced relative VE-Cadherin concentration that promotes cell-cell junction dynamics and JAIL formation. These observations nicely correlate with our own data showing the sprouting front as the area of preferential nuclear YAP/TAZ. Furthermore, we did not find junctional localisation of YAP/TAZ at the sprouting front, pointing to less sequestration of YAP/TAZ away from the nucleus by more dynamics junctions. Together, these data provide additional strength to a model in which reduced VE-Cadherin concentration at cell junctions may promote YAP and TAZ relocation to the nucleus. In zebrafish, Nakajima and colleagues (*Nakajima et al., 2017*) showed that YAP nuclear relocation correlated with lumenisation of sprouting vessels, and they attributed this to the effect of shear stress on YAP. In the mouse retina, hemodynamic fluid laws predict that vessels at the sprouting front experience very low levels of shear (*Bernabeu et al., 2014*), arguing against YAP and TAZ being activated by shear in this model. Additionally, we found no difference in the subcellular localisation of YAP or TAZ between arteries and veins, that is, vessels that experience distinct shear stress levels. However, it is possible that local and fast changes in shear stress levels are more relevant to regulate YAP and TAZ than sustained shear. In support of this idea, YAP and TAZ appear not to respond to 12 or 24 hr of laminar shear (*Wang et al., 2016*), but translocate to the nucleus after only 10 min of laminar shear (*Nakajima et al., 2017*). Finally, although VEGF, a pro-angiogenic molecule secreted by astrocytes at the avascular front, drives endothelial proliferation and migration, we found no evidence for VEGF induced YAP and TAZ nuclear translocation, although others found opposite results (*Kim et al., 2017*; *Wang et al., 2017*). Other pro-angiogenic molecules, either locally produced or blood-borne, could regulate endothelial YAP and TAZ during development; future work will help clarify these questions and how different chemical and mechanical stimuli come together to regulate YAP and TAZ.

To address the cell autonomous role of YAP and TAZ we took advantage of an endothelial specific inducible Cre to inactivate YAP and/or TAZ genetically during angiogenesis. The mild phenotype of the single mutants in comparison to the drastic phenotype of the compound mutant indicates functional redundancy in the endothelium. The compound loss of endothelial YAP and TAZ leads in the mouse retina to a decrease in the radial expansion of vessels, vascular density, branching and sprouting. This phenotype could be a consequence of a decreased number of ECs caused by a proliferation defect (*Ubezio et al., 2016*). However, our further discovery that YAP and TAZ are required to establish homogeneity in the plexus and prevent cellular aggregations suggests that endothelial YAP/TAZ signalling is not only required to provide adequate numbers of cells but is also critically involved in ensuring adequate EC distribution. We propose that endothelial YAP/TAZ operate in several mechanisms that jointly establish a balance of the right number of endothelial cells in the right place. First, endothelial YAP/TAZ drive proliferation in response to mechanical stimulation at the cell-cell junction, and not in response to VEGF. We propose that in this way endothelial YAP/TAZ provide a cell intrinsic mechanism of locally controlling cell densities, in contrast to growth factor-mediated cell proliferation instructed by the surrounding tissue. Second, endothelial YAP/TAZ increase VE-Cadherin turnover at cell-cell junctions, which we propose is essential for cells to migrate and rearrange while maintaining the endothelial barrier at the same time. This corroborates recent findings in mouse hepatocytes where YAP antagonises adherens junction stability (*Bai et al., 2016*). The authors showed that YAP regulates hepatocyte adherens junctions in response to increased actomyosin contractility by increasing myosin II light chain gene expression. Accordingly, the transcriptional, nuclear role of YAP was required for junctional regulation. Together with our observations, these findings indicate the existence of a positive feedback loop where stable junctions sequester YAP and TAZ from the nucleus, therefore maintaining less junctional turnover, while remodelling junctions allow YAP and TAZ to relocate to the nucleus where they increase VE-Cadherin turnover. Our results also suggest that a high VE-Cadherin turnover at the sprouting front is required in order to maintain junctional integrity and prevent bleedings. YAP and TAZ increase the presence of JAIL (the more dynamic type of junction in our analysis) and promote branching of actin. These results are in accordance with recent data from Sakabe and colleagues reporting that YAP and TAZ increase Cdc42 activity in lamellipodia and phosphorylation of N-WASP (*Sakabe et al.,*

*2017*), an actin binding protein that promotes branching of actin through the activation of the Arp2/3 complex.

Molecularly, how YAP and TAZ affect this complex cell behavior is not entirely clear. Our results identify that endothelial YAP/TAZ reduce the expression of Notch and BMP signaling in ECs, yet the in vitro rescue attempts show that the Notch increase is not the reason for the YAP/TAZ phenotype. This is further corroborated by in vivo data from Kim and colleagues showing that DAPT treatment fails to restore sprouting defects in YAP/TAZ mutant retinas (*Kim et al., 2017*). In contrast, reducing the increased BMP signalling experimentally corrected the permeability, and partially restored cell migration in YAP/TAZ deficient cells. This would suggest that BMP, not Notch, is a main driver of the observed cellular phenotypes. Whether the increased BMP signalling is also responsible for the phenotypes of YAP/TAZ loss of function in vivo remains to be shown. Furthermore, the full details of the involved ligands and receptors of the BMP pathway remain to be determined. Previous studies identified that BMP9/10 – Alk1 signalling is anti-angiogenic (*Larrivée et al., 2012*), while BMP2/4/6 - Alk2/Alk3 signalling is pro-angiogenic (*Lee et al., 2017*). Given the hyposprouting phenotype and reduced cell migration, we expected an increase in BMP9/10-Alk1 signalling after YAP/TAZ loss. However, the results from the panel of BMP inhibitors instead point towards a possible increase of the BMP2/4 – ALK3 pathway activity. However, given the notorious promiscuity of chemical inhibitors, and the different cellular context in which they are tested, this result should be seen as an indication at best. Further studies will need to address the nature of the BMP ligand/receptors and how the deregulated BMP signalling affects endothelial cell migration and adherens junctions in the context of YAP/TAZ deficiency.

Together, our results identify a role for YAP/TAZ in promoting endothelial cellular rearrangements through the regulation of junctional turnover and collectiveness of cell migration. Conceptually, linking stretch-induced proliferation (to balance cell numbers) with modulation of junctional turnover (to facilitate cell rearrangements) seems ideally suited to achieve the required balance of cell distribution for functional vascular patterning.

# Materials and methods

## Key resources table

| Reagent type (species) or resource | Designation | Source or reference | Identifiers | Additional information |
|---|---|---|---|---|
| strain, strain background (*Mus musculus*, C57BL/6J) | WT | The Jackson laboratories | | |
| genetic reagent (*Mus musculus*) | Yap iEC-KO, *Yap*fl/fl *Pdgfb-iCreERT2* | PMID: 27215660, PMID: 18257043 | | |
| genetic reagent (*Mus musculus*) | Taz iEC-KO, *Taz*fl/fl *Pdgfb-iCreERT2* | PMID: 27215660, PMID: 18257043 | | |
| genetic reagent (*Mus musculus*) | YapTaz iEC-KO, *Yap*fl/fl *Taz*fl/fl *Pdgfb-iCreERT2* | PMID: 27215660, PMID: 18257043 | | |
| genetic reagent (*Mus musculus*) | Taz iEC-GOF, *TAZ* S89A *EGFP Pdgfb-iCreERT2* | This paper | | Cloning information in Material and methods and *Figure 7—figure supplement 1* |
| cell line (human) | HUVEC | PromoCell and Lonza | | |
| transfected construct (human) | VE-Cadherin EGFP | PMID: 24658686 | | |
| transfected construct (human) | VE-Cadherin mEos3.2 | This paper | | Cloning information in Material and methods |
| transfected construct (human) | pCMV-flag S127A YAP | Addgene, plasmid 27370 | | |
| transfected construct (human) | 3xFLAG-pCMV5-TOPO TAZ(S89A) | Addgene, plasmid 24815 | | |
| transfected construct (human) | TEF-1 Luciferase reporter (GTIIC) | PMID: 15628970 | | |
| transfected construct (murine) | RBPj Luciferase reporter | PMID: 7566092 | | |
| transfected construct (murine) | BRE Luciferase reporter | PMID: 11729207 | | |

*Continued on next page*

*Continued*

| Reagent type (species) or resource | Designation | Source or reference | Identifiers | Additional information |
|---|---|---|---|---|
| transfected construct (human) | FOPflash Luciferase reporter | PMID: 9065401 | | |
| transfected construct (*Renilla*) | Renilla Luciferase control reporter | Promega, E2241 | | |
| antibody | Yap (rabbit polyclonal) | ThermoFisher Scientific, PA1-461894 | | Dilution 1:100 |
| antibody | Taz (rabbit polyclonal) | Sigma, HPA007415 | | Dilution 1:100 |
| antibody | Erg (goat polyclonal) | Santa Cruz Biotechnology, sc-18136 | | Dilution 1:100 |
| antibody | Erg (rabbit monoclonal) | Abcam, Ab92513 | | Dilution 1:1000 |
| antibody | VE-Cadherin (rat monoclonal) | BD Biosciences, 555289 | | Dilution 1:100 |
| antibody | VE-Cadherin (goat polyclonal) | Santa Cruz Biotechnology, sc-6458 | | Dilution 1:100 |
| antibody | VE-Cadherin 55–7 H1 - Alexa-Fluor 647 Conjugate | BD Biosciences, 561567 | | Dilution 1:200 |
| antibody | TER-119 (rat monoclonal) | R and D Systems, MAB1125 | | Dilution 1:100 |
| antibody | PECAM-1 (goat polyclonal) | R and D Systems, AF3628 | | Dilution 1:200 |
| antibody | Cleaved caspase 3 (rabbit polyclonal) | R and D Systems, AF835 | | Dilution 1:200 |
| antibody | Dll4 (goat polyclonal) | R and D Systems, AF1389 | | Dilution 1:100 |
| antibody | pSMAD1/5/8 (rabbit monoclonal) | Cell Signalling, 13820S | | Dilution 1:1000 |
| antibody | Phalloidin-Alexa-Fluor 488 | ThermoFisher Scientific, A12379 | | Dilution 1:100 |
| antibody | Ib4-Alexa-Fluor 647 Conjugate | ThermoFisher Scientific, I32450 | | Dilution 1:1000 |
| antibody | Ib4-Alexa-Fluor 488 Conjugate | ThermoFisher Scientific, I21411 | | Dilution 1:1000 |
| antibody | Ib4-Alexa-Fluor 568 Conjugate | ThermoFisher Scientific, I21412 | | Dilution 1:1000 |
| antibody | YAP 63.7 (mouse monoclonal) | Santa Cruz Biotechnology, sc-101199 | | Dilution 1:1000 |
| antibody | GAPDH (mouse monoclonal) | Millipore, MAB374 | | Dilution 1:4000 |
| sequence-based reagent | SMART pool: siGENOME siRNA YAP | Dharmacon, M-012200-00-0005 | | |
| sequence-based reagent | SMART pool: siGENOME siRNA TAZ | Dharmacon, M-016083-00-0005 | | |
| sequence-based reagent | SMART pool: siGENOME siRNA VE-Cadherin | Dharmacon, M-003641-01-0005 | | |
| sequence-based reagent | SMART pool: siGENOME siRNA Non targeting 1 | Dharmacon, D001206-13-05 | | |
| sequence-based reagent | Taqman probes for RT-qPCR | Taqman | | *Supplementary file 3* |
| commercial assay or kit | Permeability assay - Transwell membranes | Costar, 3460 | | |
| commercial assay or kit | Scratch wound assay - Culture-Insert 2 Well in μ-Dish 35 mm | Ibidi, 81176 | | |
| commercial assay or kit | Click-iT EdU Alexa Fluor 647 Imaging Kit | ThermoFisher Scientific, C10340 | | |

*Continued on next page*

*Continued*

| Reagent type (species) or resource | Designation | Source or reference | Identifiers | Additional information |
|---|---|---|---|---|
| commercial assay or kit | Propidium Iodide (PI)/ RNase Staining Solution | Cell Signalling, 4087 | | |
| commercial assay or kit | Rneasy Mini Kit | Quiagen, 74104 | | |
| commercial assay or kit | M-MLV reverse transcriptase | ThermoFisher Scientific, 28025013 | | |
| commercial assay or kit | RevertAid First Strand cDNA Synthesis Kit | ThermoFisher Scientific, K1621 | | |
| commercial assay or kit | Agilent RNA 6000 Nano Kit | Agilent, 5067–1511 | | |
| commercial assay or kit | GeneChip Human Gene 2.0 ST Array | ThermoFisher Scientific, 902113 | | |
| chemical compound, drug | 250 kDa FITC Dextran | Sigma, FD250 | | |
| chemical compound, drug | Lipofectamine 2000 | ThermoFisher Scientific, 11668019 | | |
| chemical compound, drug | Dharmafect 1 transfection reagent | Dharmacon, T-2001 | | |
| chemical compound, drug | Polybrene | Santa Cruz, sc-134220 | | |
| chemical compound, drug | Hydroxytamoxifen | Sigma, 7904 | | |
| chemical compound, drug | DBZ | Cayman chemicals 14627 | | |
| chemical compound, drug | Recombinant-hGremlin | R and D Systems, 5190-GR | | |
| chemical compound, drug | Recombinant-hEndoglin | R and D Systems, 1097-EN | | |
| chemical compound, drug | LDN-193189 | Cayman chemicals, 19396 | | |
| chemical compound, drug | K02288 | Cayman chemicals, 16678 | | |
| chemical compound, drug | Recombinant hAlk1fc | R and D Systems, 370-AL-100 | | |
| chemical compound, drug | VEGF-165 (murine) | Prepotech, 450–32 | | |
| software, algorithm | FIJI | FIJI | | |
| software, algorithm | Cytoplasm to nucleus translocation assay | Cell Profiler, adapted from PMID: 17076895 | | |
| software, algorithm | Mouse retina regularity script | This paper | | *Source code 1* |
| software, algorithm | VE-Cadherin turnover analysis script | This paper | | *Source code 2* |
| software, algorithm | Patching script | This paper | | *Source code 3* |
| software, algorithm | Cell coordination analysis script | This paper | | *Source code 4* |
| software, algorithm | Dll4 gradient analysis script | This paper | | *Source code 5* |

## Mice and treatments

For loss of function experiments the following mouse strains were used: *Yap* [fl/fl] and *Taz* [fl/fl] (*Gruber et al., 2016*), *Pdgfb-iCreERT2* (*Claxton et al., 2008*). A detailed description of the knock-in mice overexpressing the TAZ gain-of-function allele will be provided elsewhere. Briefly, *3xFLAG-TAZ$^{S89A}$-IRES-nEGFP* with a preceding floxed STOP cassette was knocked into the *Rosa26* locus. Cre-mediated removal of the STOP sequence leads to CAG promoter-driven expression of 3xFLAG-tagged TAZ$^{S89A}$ as well as of nuclear-localized enhanced green fluorescence protein (nEGFP). The allele was kept heterozygous in the experimental studies and was developed together with genOway.

Mice were maintained at the London Research Institute and at the Max Delbruck Center for Molecular Medicine (loss of function mice) and at the Max Planck Institute for Heart and Lung Research (gain of function mice) under standard husbandry conditions. To induce Cre-mediated

recombination 4-hydroxytamoxifen (Sigma, 7904) was injected intraperitoneally (IP) (20 µL/g of 1 mg/mL solution) at postnatal day 1 and day 3 and eyes were collected at P6. In all loss and gain of function experiments control animals were littermate animals without Cre expression. Male and female mice were used for the analysis.

For endothelial cell proliferation assessment in the retina, mouse pups were injected IP 2 hours before culling with 20 uL/g of EdU solution (0.5 mg/mL; Thermo Fischer Scientific, C10340).

## Cell culture

HUVECs from pooled donors (PromoCell) were cultured in EGM2-Bulletkit without antibiotics (Lonza) and used until passage 6. For YAP and TAZ gain of function experiments HUVECs were obtained from Lonza, cultured in endothelial basal medium (Lonza) supplemented with hydrocortisone (1 µg ml$^{-1}$), bovine brain extract (12 µg ml$^{-1}$), gentamicin (50 µg ml−1), amphotericin B (50 ng ml$^{-1}$), epidermal growth factor (10 ng ml$^{-1}$) and 10% fetal bovine serum (Life Technologies) and used until passage 4. The manufacturers authenticated the identity of HUVECs by flow cytometry for cell-type specific markers (vWF, CD31, CD105) and by functional analysis (cells positive for acetylated low density lipoprotein uptake). All cells were tested negative for mycoplasma.

For knockdown experiments, HUVECs were transfected with SMARTpool: siGENOME siRNAs purchased from Dharmacon (Yap #M-012200-00-0005, Taz #M-016083-00-0005, VE-Cadherin #M-003641-01-0005 and non-targeting siRNA Pool 1 #D001206-13-05). Briefly, subconfluent (70–80%) HUVECs were transfected with 25 nM siRNA using Dharmafect 1 transfection reagent following the protocol from the manufacturer; transfection media was removed after 24 hr and experiments were routinely performed on the third day after transfection.

To activate YAP and TAZ signalling in ECs, FLAG-YAP$^{S127A}$- or 3x-FLAG-TAZ$^{S89A}$-encoding adenoviruses were generated in the adenoviral type five backbone lacking the E1/E3 genes (Vector Biolabs). GFP-encoding adenoviruses were used as a control. Infections were carried out by incubating sub-confluent HUVECs (70–80%) with starvation media (EBM containing 0.1% BSA) for 4 hr followed by the addition of adenoviral particles and polybrene (Santa Cruz). After 4 hr, HUVECs were washed with Hanks Buffer for at least five times and then cultured in complete EBM media with 10% FCS and supplements overnight. All experiments were performed 24 hr post transduction.

## Immunofluorescence staining

To perform retina immunofluorescence, eyes were collected from postnatal day six mice and fixed in 4% PFA in PBS for 1 hr at 4C. Retinas were dissected in PBS and permeabilised/blocked for 1 hr at room temperature in 1% BSA, 2% FBS, 0.5% Triton X100, 0.01% Na deoxycholate and 0,02% Na Azide in PBS. Primary and secondary antibodies were incubated overnight at 4C and for 2 hr at room temperature, respectively, both in 1:1 PBS: blocking buffer. Isolectin staining was performed overnight at 4C in Pblec after retinas were equilibrated for 1 hr in Pblec at room temperature. Retinas were post-stained fixed in 2% PFA in PBS for 10 min. To mount the samples Vectashield mounting medium. (Vector Labs, H1000) or ProLong Gold (Thermo Fisher Scientific) was used. Imaging was done by laser scanning confocal microscopy (Carl Zeiss LSM700, LSM780 and Leica TCS SP8). Processing of samples was carried out in tissues from littermates under the same conditions.

For immunofluorescence in HUVECs, cells were grown in #1.5 coverslips coated with poly-lysine and gelatin 0.2%. At the end of the experiment cells were fixed in 4% PFA for 10 min, permeabilised in 0.3% Triton-X100 in blocking buffer for 5 min and blocked in 1% BSA 20 mM Glycine in PBS for 30 min. Primary and secondary antibodies were incubated for 2 and 1 hr, respectively, in blocking buffer. Nuclei labeling was performed by incubating cells with DAPI for 5 min (Life technologies, D1306).

A list of the primary antibodies used can be found in *Supplementary file 2*.

## Image analysis

Analysis of radial expansion, capillary density, branching frequency, proliferating ECs, apoptosis and sprouting numbers was done using Fiji (*Schindelin et al., 2012*). Radial expansion corresponds to the mean distance from the optic nerve to the sprouting front (eight measurements in tilescans of two whole retinas per animal). Capillary density corresponds to the vessel area (measured by thresholding IB4 signal) divided by the field of view area (6–8 images of (425 µm)$^2$ between artery and vein

per animal). Branching frequency was measured by manually counting all branching points in a field of view (4–5 images of (200 µm)$^2$ between artery and vein per animal). The plexus regularity was assessed through the standard deviation of the size and the circularity of the vascular loops in the plexus (using same images as for analysis of capillary density). Vascular loops were segmented by thresholding the IB4 signal to avoid artifacts we excluded loops with a size smaller than 86 um̃2 for the analysis. Endothelial proliferation was measured by manually counting the number of EdU positive endothelial nuclei (ERG positive) and dividing by the vessel area (measured by thresholding IB4 signal) (4 images of (425 µm)$^2$ containing the sprouting front and localized on top of arteries per animal). Apoptosis was measured manually by counting the number of cleaved caspase 3 positive figures and dividing by the vessel area (measured by thresholding IB4 signal) (tilescan of one whole retina per animal). The number of sprouts was measured manually (3 images of 425 × 850 µm of the sprouting front per animal). To quantify DLL4 intensity the outline of the sprouting front and the position of the arteries were manually defined using IB4 staining. Vessels were segmented by thresholding the IB4 staining in Fiji. Then, DLL4 intensity inside the vasculature was normalised with the average DLL4 intensity outside of the vasculature. Subsequently, for every pixel inside the vasculature (excluding the arteries) the distance to the sprouting front was calculated. The normalised DLL4 values within each bin were averaged (15 µm bins from 0 to 500 µm). For each retina quarter a curve was obtained, and the average and SEM of these curves was shown in the graph (one retina quarter was used per animal). To quantify pSMAD1/5/8 status the number of pSMAD1/5/8 positive endothelial nuclei was manually counted and dividing by the total number of endothelial nuclei (defined by being ERG positive) (3 images of (225 µm)$^2$ containing the sprouting front were used per animal).

To analyse YAP/TAZ subcellular localisation in HUVECs we adapted a previously existing cytoplasm-to-nucleus translocation assay pipeline from Cell Profiler (*Carpenter et al., 2006*). Briefly, YAP or TAZ staining intensity was measured both inside the nucleus of the cell and in a 12 pixels wide ring of cytoplasm grown radially from the nucleus. The nucleus localisation was determined using a DAPI mask.

Cell junction morphology analysis was done in confluent monolayers of HUVECs stained for VE-Cadherin. Five morphological categories were defined: straight, thick, thick to reticular, reticular and fingers. We acquired 5 images of (160 µm)$^2$ per condition per experiment, divided each image in (16 µm)$^2$ patches, and randomly grouped these patches. The classification into categories was done manually and blindly for the condition.

To analyse cell coordination we used confluent cells labelled for DAPI. The nuclei were automatically segmented using a customized Python algorithm relying on the Scikit Image Library. By fitting an ellipse to each nucleus we obtained its major and minor axis, and the angle of the major axis with the x-axis of the image was assigned to the nucleus as its orientation. This way each nucleus in the images was assigned a position given by its midpoint and an orientation. Next we analyzed the average alignment of the nuclei of two cells depending on their distance. As the nuclei don't have a directionality (i.e. they are nematics as opposed to vectors), the angles between two nuclei range from 0 corresponding to the nuclei being parallel, to $\pi/2$ corresponding to them spanning a right angle. For any two cells in each image we calculated the angle and the Euclidean distance between them, and then we binned the cells depending on their distance. We introduced a parameter called 'alignment' which is one if all cells are perfectly aligned and 0 for a completely random distribution of cell orientations.

## Live imaging of VE-Cadherin-EGFP

24 hr after siRNA transfection, knockdown HUVECs were transduced with VE-cadherin-EGFP adenovirus as described before (*Bentley et al., 2014*). Briefly, cells were incubated with the virus for 24 hr and then washed three times to remove viral particles. Cells were replated onto 2-well LabTek chambered coverslips (Nunc) coated with 10 ug/mL Fibronectin (Sigma, F1141). Imaging was performed 48 hr post-transduction. Cells were imaged at 37°C under 5% $CO_2$ on LSM 780 (Zeiss) using a Plan-Apochromat 63x/1.4 oil objective. Images were acquired at a 260 s time frame.

## VEGF treatment and YAP/TAZ staining

Confluent HUVECs were maintained in VEGF free media for 24 hr. VEGF treatment was then performed for 30 min, 1 hr and 3 hr with 0 or 40 ng/mL of VEGF-165 (PrepoTech, 450–32).

Immunofluorescence staining and analysis of YAP and TAZ subcellular localisation was performed as above described.

## VEGF treatment and proliferation assessment

Knockdown HUVECs were maintained in VEGF free media for 24 hr. VEGF treatment was then performed for 24 hr with 0 ng/mL, 40 ng/mL, 200 ng/mL or 1000 ng/mL of VEGF-165. Cells were pelleted, ressuspended in 90% cold Methanol and stored at −20C° before further processing. Cells were then ressuspended in Propidium Iodide/RNase staining solution (Cell signaling, 4087) for 30 min before cell cycle analysis by flow cytometry (LSRII, BD). Data was analysed using BD FACSDiva software.

## Mechanical stretch application and proliferation assessment

HUVECs were plated on collagen I - 0.2% gelatine-coated Bioflex plates (BF-3001C, Flexcell International Corporation). Gene knockdown was preformed as previously described. Cells were incubated in transfection media for 24 hr, and allowed to recover in fresh complete media for 4 hr. Afterwards cells were incubated for 24 hr in serum starvation media (0,1%BSA in EBM2 pure media) to form a confluent, quiescent monolayer. Cyclic stretch (0.25 Hz, 15% elongation) was then applied for 24 hr using a Flexcell FX-5000 Tension System. Control cells were placed in the same incubator but not on the Flexcell device (static conditions). EdU pulsing was performed after 20 hr of the 24 hr stretch period. At the end of the experiment cells were fixed in 4% PFA and EdU staining was performed according to the manufacturer's protocol (Click-It EdU C10340 Life Technologies). Nuclei were labelled with DAPI. Three regions of interested were acquired per sample in a Carl Zeiss LSM700 scanning confocal microscopes (Zeiss, Germany). Quantification of proliferation was done using a CellProfiler pipeline. Percentage of S phase cells was determined as percentage of EdU positive nuclei over the total number of nuclei.

## Permeability assay

24 hr after siRNA transfection cells were re-plated into fibronectin coated Transwell membranes (Costar 3460) at confluence and incubated for two more days to stabilize cell junctions. On the third day after transfection 0.5 mg/mL of 250 kDa FITC Dextran in cell media (Sigma FD250) was added to the top well. Fluorescence on the bottom well was measured after 6 hr in a Gemini XPS fluorescent plate reader.

## Pulse chase VE-Cadherin experiment for quantification of low, intermediate and high turnover junctions

Cells were labelled live with a non-blocking monoclonal antibody directed against extracellular VE-Cadherin and directly coupled with Alexa-Fluor647 (BD Pharmingen, #561567, 1:200) for 30 min. Cells were then washed 2x with PBS and incubated with complete media for additional 2 hr. Cells were fixed with 4% PFA and stained for VE-Cadherin (Santa Cruz Biotechnology, #6458, 1:200) with a secondary antibody coupled with Alexa-Fluor-488. 5 (160μm)$^2$ images per condition per experiment were acquired in a Carl Zeiss LSM700 confocal laser scanning microscope using the same acquisition settings. Max projection of z stack and merging of channels was done in Fiji. Images were divided in (16 μm)$^2$ patches and the patches were randomly grouped. Patches were classified into a morphological category and into low, intermediate or high turnover categories, manually and blindly for the condition.

## VE-cadherin mEos3.2 cloning

mEos3.2 cDNA (*Zhang et al., 2012*) was cloned downstream of full-length human VE-cadherin with a short linker (ARDPPV) and inserted into pAc-GFP-N1 backbone (Clontech) using NEBbuilder HiFi Assembly mix (NEB).

## Fluorescent loss after photoconversion experiments

HUVECs double-transfected with YAP/TAZ or scrambled siRNAs and pN1-CMV-VE-cadherin-mEos3.2 were cultured to confluency in 2-well LabTek chambered coverslips (Nunc) coated with 10 ug/mL Fibronectin (Sigma, F1141) in EGM (Promocell) supplemented with EGM2 bulletkit (Lonza).

Cells were imaged at 37°C under 5% $CO_2$ on LSM 780 (Zeiss) equipped with Definite Focus stabilizer. Imaging was performed using the 488 nm (green mEos3.2 component) and the 561 nm (red component) lasers using Plan-Apochromat 63x/1.4 oil objective, 0.26 $\times$ 0.26 µm pixel size and 5.09 µs pixel dwell time, 16-bit image depth. A circular region of interest (ROI) of 21 µm$^2$ area was selected on straight junctions and photoconverted using the 405 nm laser. Mean fluorescence intensity in the ROI was monitored in the red channel for 15 min with 10 s resolution, while the movement of the junction was followed in the green channel.

Background signal in each frame was estimated by measuring mean intensity in non-photoconverted region and subtracted from the fluorescence-loss curves. The curves were further corrected for bleaching, using parameters estimated from fixed cells. The curves were normalized between the mean intensity in the frames before photoconversion and fluorescence intensity measured immediately after photoconversion. Normalized curves were smoothed using the moving average method; the half-time of redistribution and immobile fraction of VE-cadherin were estimated directly from the plots. Only junctions which did not substantially move or remodel during the observation time were considered for analysis. Analysis was performed using Fiji and Matlab (Mathworks).

## Scratch wound assay

24 hr after siRNA transfection cells were re-plated into a scratch wound assay device (IBIDI). On the following day a cell free gap of 500 µm was created by removing the insert of the device. Images were taken immediately after removing the insert (0 hr) and after 16 hr. The cell free area was measured in Fiji and used to calculate the percentage of wound closure at 16 hr.

## RNA extraction and quantitative real time-polymerase chain reaction

RNA was extracted using the RNeasy Mini Kit (Qiagen) according to the manufacturer's instructions. For HUVECs transfected with adenoviruses carrying *YAP* and *TAZ* gain of function mutations, 2 µg of total RNA were reverse transcribed to cDNA using M-MLV reverse transcriptase (ThermoFisher Scientific). For HUVECs transfected with siRNAs 90 ng of RNA were reverse transcribed using RevertAid First Strand cDNA Synthesis Kit (ThermoFisher Scientific). qRT-PCR was performed using TaqMan reagents and probes (Applied Biosystems) (listed in *Supplementary file 3*). qRT-PCR reactions were run on a StepOnePlus real-time PCR instrument (ThermoFisher Scientific) or Quant Studio 6 Flex (Applied Biosystems) and expression levels were normalised to human *ACTB* or human *HPRT1* using the 2deltaCT method.

## Western blot

Protein was extracted from HUVECs using M-PER protein extraction reagent with Halt Protease and Phosphatase inhibitors (Pierce). Proteins concentration was assessed using a BCA protein assay kit (Pierce). Proteins were separated by SDS–PAGE and blotted onto nitrocellulose membranes (Bio-Rad). Membranes were probed with specific primary antibodies and then with peroxidase-conjugated secondary antibodies. The following antibodies were used: YAP 63.7 (Santa Cruz Biotechnology, sc-101199, 1:1000), GAPDH (Millipore, MAB374, 1:4000). The bands were visualized by chemiluminescence using an ECL detection kit (GE Heathcare) and a My ECL Imager (Thermo Scientific).

## Dual luciferase reporter assay

Renilla-luciferase reporter assays for TEF-1 (*Mahoney et al., 2005*), RBPj (*Jarriault et al., 1995*), BRE (*Korchynskyi and ten Dijke, 2002*; *Fritzmann et al., 2009*) and FOPflash (*Korinek et al., 1997*)-Luciferase promoter activity were performed as follows: 48 hr after gene knockdown by siRNA HUVECs were cotransfected with 600 ng of Luciferase reporter gene construct and 300 ng of pRL-TK (Promega) using Lipofectamine2000 and incubated for 4 hr. Cell extracts were prepared 72 hr post siRNA transfection and 24 hr post Luciferase reporter transfection, and luciferase activity was measured using a dual luciferase system as described (*Hampf and Gossen, 2006*). Experiments were carried out in duplicates and results were normalized to the correspondent FOPflash/Renilla measurement.

## Microarray and gene set enrichment analysis

Microarray studies were performed as described(*Murtaugh et al., 2003*). In brief, total RNA was extracted from HUVECs using the RNeasy kit (Qiagen) and RNA quality assessed with the 6000 nano kit and an Agilent Bioanalyser. RNA was labelled according to the Affymetrix Whole Transcript Sense Target Labeling protocol. Affymetrix GeneChip Human Gene 2.0 ST arrays were hybridized and scanned using Affymetrix protocols. Data were analysed using the Affymetrix expression console using the RMA algorithm; statistical analysis was done using DNAStar Arraystar 11. Heat maps of gene signatures were plotted using RStudio, Inc.

## Notch and BMP inhibition experiments

A list of the reagents used, together with duration of treatment, can be found in *Supplementary file 1*.

## Statistical analysis

Statistical analyses were performed using GraphPad Prism software and *p* value was determined using unpaired Student *t*-test between the control and the knockout/knockdown/ gain of function condition. Statistical significance was considered for p<0.05. Values shown are mean and standard deviation was used as the dispersion measure. Biological replicates refer to individual mice for in vivo experiments and different wells for in vitro cell culture experiments; independent experiments refer to experiments done in different days; technical replicates refer to repeated measurements taken from the same sample, both for in vivo and in vitro. Exclusion of outliers was done using 'Robust regression and Outlier removal' from GraphPad Prism software, with a coefficient Q of 1%. A statistical method of sample size calculation was not used during study design. For in vivo experiments, we used an average of 6 animals per experiment, from different litters, with a minimum of 3 (detailed number of animals used in figure legends and source data). For in vitro experiments, we did a minimum of 3 independent experiments (detailed number of experiments in figure legends and source data). When technically possible the investigators were blind to the genotype of the animal or cell culture condition during sample processing and data analysis.

## Acknowledgements

We thank members of the Vascular Biology (Berlin) and Vascular Patterning (VIB – Leuven) Laboratories for helpful discussions. We thank the Cancer Research UK - London Research Institute and the Max Delbrück Center for Molecular Medicine Animal Facilities for animal care and technical support. We thank Dr. Axel Behrens for kindly providing the *Taz^{fl/fl}* mice. We thank Dr. Walter Birchmeier and Dr. Daniel Besser for providing the Normalizer and BRE-luc reporter, Dr. Eric Sahai and Dr. Nic Tapon for providing the TEF1-reporter, and Dr. Michael Gotthardt and Dr. Michael Radke for access to the Flexcell Tension System and technical assistance. We thank Dr. Dietmar Vestweber for providing the VE-Cadherin-EGFP adenovirus. We specially thank Dr. Veronique Gebala, Dr. Andre Rosa and Dr. Baptiste Coxam for helpful comments on the manuscript. FN was financially supported by the Fundação para a Ciência e a Tecnologia (FCT), CRUK-CRICK and the MDC. ACV, AKB and EBK were supported by the DZHK (German Centre for Cardiovascular Research), AS was supported by the EMBO (European Molecular Biology Organization), JRC was supported by the FCT. CAF is supported by the FCT, EC-ERC Starting Grant, Portugal2020 program. MP is supported by the Max Planck Society, the ERC Starting Grant ANGIOMET, the Deutsche Forschungsgemeinschaft, the Excellence Cluster Cardiopulmonary System, the LOEWE grant Ub-Net, the DZHK, the Stiftung Charité and the EMBO Young Investigator Program. HG is supported by the DZHK and ERC Consolidator Grant Reshape 311719.

## Additional information

### Competing interests

Holger Gerhardt: Reviewing editor, *eLife*. The other authors declare that no competing interests exist.

## Funding

| Funder | Grant reference number | Author |
|---|---|---|
| H2020 European Research Council | Consolidator Grant Reshape 311719 | Holger Gerhardt |
| Fundação para a Ciência e a Tecnologia | SFRH/BD/51287/2010 | Filipa Neto |
| European Molecular Biology Organization | Long-Term Fellowship ALTF 1625-2014 | Anna Szymborska |
| Deutsches Zentrum für Herz-Kreislaufforschung | REMODEL | Alexandra Klaus-Bergmann Anne-Clémence Vion Eireen Bartels-Klein Holger Gerhardt |
| H2020 European Research Council | TWINN-2015 ReTuBi-692322 | Claudio A Franco |
| Fundação para a Ciência e a Tecnologia | SFRH/BD/52224/2013 | Joana R Carvalho |
| Fundação para a Ciência e a Tecnologia | FCT Investigator IF/00412/2012 | Claudio A Franco |
| H2020 European Research Council | EC-ERC Starting Grant AXIAL.EC-679368 | Claudio A Franco |
| Fundação para a Ciência e a Tecnologia | EXPL/BEX-BCM/2258/2013 | Claudio A Franco |
| H2020 European Research Council | Starting Grant ANGIOMET (311546) | Michael Potente |
| Deutsche Forschungsgemeinschaft | SFB 834 | Michael Potente |
| Excellence Cluster Cardiopulmonary System | EXC 147/1 | Michael Potente |
| LOEWE Research Initiatives Network | Ub-Net | Michael Potente |
| Stiftung Charité | | Michael Potente |
| European Molecular Biology Organization | Young Investigator Programme | Michael Potente |
| Portugal2020 Program | LISBOA-01-0145-FEDER-00739 | Claudio A Franco |

The funders had no role in study design, data collection and interpretation, or the decision to submit the work for publication.

## Author contributions

Filipa Neto, Conceptualization, Resources, Data curation, Formal analysis, Funding acquisition, Validation, Investigation, Visualization, Methodology, Writing—original draft, Project administration, Writing—review and editing; Alexandra Klaus-Bergmann, Data curation, Formal analysis, Funding acquisition, Validation, Investigation, Methodology, Writing—review and editing; Yu Ting Ong, Resources, Data curation, Formal analysis, Validation, Investigation, Visualization, Methodology, Writing—review and editing; Silvanus Alt, Data curation, Formal analysis; Anne-Clémence Vion, Data curation, Formal analysis, Funding acquisition, Validation, Visualization, Methodology, Writing—review and editing; Anna Szymborska, Software, Formal analysis, Investigation, Writing—review and editing; Joana R Carvalho, Formal analysis, Investigation, Writing—review and editing; Irene Hollfinger, Eireen Bartels-Klein, Investigation; Claudio A Franco, Writing—review and editing; Michael Potente, Conceptualization, Resources, Supervision, Funding acquisition, Visualization, Methodology, Project administration, Writing—review and editing; Holger Gerhardt, Conceptualization, Resources, Data curation, Supervision, Funding acquisition, Writing—original draft, Project administration, Writing—review and editing

### Author ORCIDs

Filipa Neto (iD) http://orcid.org/0000-0001-8382-2269
Yu Ting Ong (iD) http://orcid.org/0000-0003-3407-2515
Anne-Clémence Vion (iD) http://orcid.org/0000-0002-2788-2512
Claudio A Franco (iD) http://orcid.org/0000-0002-2861-3883
Holger Gerhardt (iD) http://orcid.org/0000-0002-3030-0384

### Ethics

Animal experimentation: All procedures involving handling of living animals were performed in accordance with: the United Kingdom's Home Office Animal Act 1986 under the authority of project licence PPL 80/2391 (experiments done in LRI-CRUK); the German animal protection law with approval by the regional offices for health and social services LaGeSo, under the animal licence IC113 G 0117/15 (experiments done in MDC); institutional guidelines and laws, following protocols approved by local animal ethics committees and authorities (B2/1061; Regierungspraesidium Darmstadt) (experiments done at the MPI for Heart and Lung Research).

### Decision letter and Author response

Decision letter https://doi.org/10.7554/eLife.31037.042
Author response https://doi.org/10.7554/eLife.31037.043

## Additional files

### Supplementary files

• Source code 1. Mouse retina regularity script. Determines the regularity of the gaps in the mouse retina vasculature Used in *Figure 2r,K,L*. Written in Python.
DOI: https://doi.org/10.7554/eLife.31037.028

• Source code 2. VE-Cadherin turnover analysis script. Used in *Figure 5K,L*. Written in Python.
DOI: https://doi.org/10.7554/eLife.31037.029

• Source code 3. Patching script. Used in *Figure 5F,K,L* and *Figure 8I*. Written in Python.
DOI: https://doi.org/10.7554/eLife.31037.030

• Source code 4. Cell coordination analysis script. Segments images of DAPI stained cell nuclei in a confluent monolayer and assesses the alignment between cells as a function of their distance. Used in *Figure 6N,O*. Written in Python.
DOI: https://doi.org/10.7554/eLife.31037.031

• Source code 5. Dll4 gradient analysis script. Analyses Dll4 intensity in the mouse retina as a function of the distance to the sprouting front. Used in *Figure 7—figure supplement 4*. Written in Python.
DOI: https://doi.org/10.7554/eLife.31037.032

• Supplementary file 1. List of reagents used to manipulate Notch and BMP signaling in cell culture.
DOI: https://doi.org/10.7554/eLife.31037.033

• Supplementary file 2. List of primary antibodies and dyes used.
DOI: https://doi.org/10.7554/eLife.31037.034

• Supplementary file 3. List of the TaqMan primers (Applied Biosystems) used.
DOI: https://doi.org/10.7554/eLife.31037.035

• Transparent reporting form
DOI: https://doi.org/10.7554/eLife.31037.036

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
