## [Decision Letter]

Thank you for submitting your article "YAP and TAZ regulate adherens junction dynamics and endothelial cell distribution during vascular development" for consideration by *eLife*. Your article has been reviewed by three peer reviewers, one of whom, Reinhard Fässler (Reviewer #1), is a member of our Board of Reviewing Editors, and the evaluation has been overseen by Fiona Watt as the Senior Editor. The following individual involved in review of your submission has agreed to reveal their identity: Guido Serini (Reviewer #2).

Summary:

The authors report that YAP/TAZ activity is mechanically regulated in endothelial cells. This regulation of YAP/TAZ in turn controls EC proliferation and migration, VE-cadherin turnover, and Notch and BMP signalling. The reviewers spent quite a bit of time reviewing and then discussing this paper, in part because some of the findings were already published (Kim et al. JCI). However, the reviewers concluded that work on the role of YAP/TAZ in vascular biology is important, particularly since it was carried out in vivo (with loss and gain of function models) and reaches slightly different conclusions than the published work by Kim et al. Based on this discussion the Reviewing Editor has drafted this decision to help prepare a revised submission. We hope it will be possible to submit the revised version within two to three months.

Essential revisions:

1) It is stated in the manuscript that "VEGF treatment did not alter the subcellular localisation of YAP and TAZ in HUVECs (Figure 3—figure supplement 2), suggesting that VEGF is not a primary regulator of their activity". The lack of a link between VEGF and YAP/TAZ activation could be due to the experimental design. The Materials and methods say that "Confluent HUVECs were maintained in VEGF free media for 24h. VEGF treatment was then performed for 6h with 0ng/mL, 4ng/mL, 20ng/mL or 100ng/mL of VEGF-165".

The VEGF treatment is probably too long to see an effect on YAP/TAZ translocation. It was previously reported that 30 min treatment with a prototypical growth factor, such as EGF, induces YAP nuclear accumulation in MCF-10A cells (Fan et al., PNAS 2013; 110: 2569-2574). Furthermore, Kim et al. treated HUVECs for 30 min, as did Wang et al. in their paper (Developmental Cell, 2017). Hence, we propose to repeat the VEGF treatment for shorter periods (30 min, 1hr, 3hrs), quantify nuclear translocation of YAP/TAZ by co-staining with a nuclear marker, and compare and discuss the findings against the published results.

2) The VE-cadherin turnover needs to be better analysed. There is a body of evidence in the literature demonstrating that VE-cadherin internalization is key to increase vascular permeability in vitro and in vivo (Gavard and Gutkind, Nat. Cell Biol. 2006, 8:1223-1234; Gavard et al., Dev. Cell 2008, 14:25-36; Dwyer et al., Oncogene 2011, 30:190-200; Orsenigo et al., Nat. Commun. 2012, 3:1208; Yoshioka et al., Nat. Med. 2012, 18:1560-1569; Wessel, F. et al., Nat. Immunol. 2014, 15:223-230; Dorland et al., Nat. Commun. 2016, 7:12210; Di Russo et al., EMBO J. 2017, 36: 183-201). The observation that YAP/TAZ knockdown would increase endothelial cell permeability and at the same time reduce VE-cadherin endocytosis appears difficult to be reconciled in light of the above mentioned papers.

The problem is that pulse-chase experiments provide only a semi-quantitative analysis of the VE-cadherin turnover at cell-cell contacts rather than a quantitative analysis VE-cadherin endocytosis or recycling. FRAP experiments may be a more precise quantitative approach to evaluate VE-cadherin turnover at cell-cell junctions in YAP/TAZ-depleted HUVECs. We also advise to characterise the colocalization of VE-cadherin with markers of early and/or late endosomes, such as EEA1 or LAMP1, respectively.

3) We find the YAP/TAZ link interesting, although such a link has recently been proposed by Totaro et al. (2017). The role of Notch and its epistatic relationship with YAP and/or TAZ should be somehow tackled experimentally, at least in vitro, and demonstrated not in all but at least in few key experiments. There are good reagents to manipulate Notch in cell culture and to couple it as downstream effector of YAP/TAZ (at least partially). The authors should also consider to test whether Notch serves as a mechanical mediator of (some) YAP/TAZ biology for endothelial cells.

[Editors' note: further revisions were requested prior to acceptance, as described below.]

Thank you for resubmitting your work entitled "YAP and TAZ regulate adherens junction dynamics and endothelial cell distribution during vascular development" for further consideration at *eLife*. Your revised article has been favorably evaluated by Fiona Watt (Senior editor) and three reviewers, one of whom is a member of our Board of Reviewing Editors.

The manuscript has been improved but there are some remaining issues that need to be addressed before acceptance. In light of a recent paper published by the Schnittler lab in Nat. Commun., 2017, we would appreciate if you make the following changes in the text:

1) Explain in the Introduction section that two classes of VE-cadherin-based adhesions exist: adherent junctions (characterizing endothelial cells of mature and already formed blood vessels) and JAILs (typical of endothelial cells of newly forming and very dynamic angiogenic blood vessels).

2) Adopt the Schnittler nomenclature throughout the manuscript by re-defining their "reticular junctions" as "large JAILs" and "thick to reticular junctions" as "small JAILs" (as also mentioned by authors in subsection “YAP/TAZ regulate adherens junction morphology and stability”). This will allow readers to more easily compare the complementary concepts exposed in these manuscripts.

3) Mention that the preferential nuclear YAP/TAZ localization in endothelial cells of the sprouting front of the developing retinal vasculature nicely fit with the observation reported by Cao et al. (Nat. Commun., 2017, 8:2210). Indeed, in Figure 1 Cao et al. show that endothelial cells of blood vessels localized at the sprouting front of the developing retina are elongated and display a reduced relative VE-cadherin concentration that promotes cell-to-cell junction dynamics and JAIL formation. Altogether, these data would provide additional strength to a model in which reduction of VE cadherin concentration at endothelial cell-to-cell junctions may promote YAP and TAZ relocation to the nucleus.

---

## [Author Response]

Essential revisions:1) It is stated in the manuscript that "VEGF treatment did not alter the subcellular localisation of YAP and TAZ in HUVECs (Figure 3—figure supplement 2), suggesting that VEGF is not a primary regulator of their activity". The lack of a link between VEGF and YAP/TAZ activation could be due to the experimental design. The Materials and methods say that "Confluent HUVECs were maintained in VEGF free media for 24h. VEGF treatment was then performed for 6h with 0ng/mL, 4ng/mL, 20ng/mL or 100ng/mL of VEGF-165".The VEGF treatment is probably too long to see an effect on YAP/TAZ translocation. It was previously reported that 30 min treatment with a prototypical growth factor, such as EGF, induces YAP nuclear accumulation in MCF-10A cells (Fan et al., PNAS 2013; 110: 2569-2574). Furthermore, Kim et al. treated HUVECs for 30 min, as did Wang et al. in their paper (Developmental Cell, 2017). Hence, we propose to repeat the VEGF treatment for shorter periods (30 min, 1hr, 3hrs), quantify nuclear translocation of YAP/TAZ by co-staining with a nuclear marker, and compare and discuss the findings against the published results.

As our VEGF experiments were conflicting with the previously published results by Kim et al. (1) and Wang et al. (2), we repeated the experiments using the shorter periods of treatment as suggested by the reviewers and quantified translocation of YAP/TAZ by co-staining with the nuclear marker DAPI. We also noticed on reviewing our experiments that we were not using 0-4-20-100ng/mL as stated, but instead 0-40-200-1000ng/mL of VEGF-165, which we corrected in the manuscript accordingly. We therefore repeated the staining for YAP and TAZ using 40ng/mL VEGF-165 concentration, as this was most equivalent to the concentration used in the referred papers (50ng/mL), and evaluated YAP/TAZ nuclear to cytoplasmic ratio after 30 min, 1h and 3h treatment. The results showed no differences upon VEGF treatment at any time point (new Figure 3—figure supplement 2).

In Wang et al. (2) cells were cultured routinely in 10% FBS, and starved overnight in serum free media before VEGF stimulation. Because our starvation media contained 2% FBS, we repeated the experiment doing 24h of serum starvation (EBM2 pure – without added growth factors and FBS – supplemented with 0.1% BSA) (Author response image 1). Also in these conditions quantification revealed no activation of YAP/TAZ upon VEGF treatment.

**Author response image 1. respfig1:** VEGF treatment does not affect YAP and TAZ subcellular localisation. Confluent HUVECs were starved in FBS free media (EBM2 0.1% BSA) and treated with 40ng/mL of VEGF for 30 min (B, F), 1h (C,G) and 3h (D, H) or control (A, E) and stained for YAP (**A-D**) or TAZ (**E-H**) and DAPI (not shown). **I,J,** Quantification of nuclear to cytoplasmic ratio of YAP (**I**) or TAZ (**J**) with VEGF treatment. Nuclear/cytoplasmic ratio > 1, YAP/TAZ nuclear; nuclear/cytoplasmic < 1, YAP/TAZ cytoplasmic. At least 200 cells quantified.

The difference that we see in our experiments did not seem to be explained by the type of cells, confluency, culture conditions or starvation (Kim et al. (1) also used HUVECs – confluent conditions – EBM2 Lonza media, cells cultured in gelatine – starvation 24h pure EBM2). However, we cannot exclude differences due to the nature or source of VEGF used. We used a commercial VEGF (murine VEGF165 from Prepotech) (and confirmed it’s bioactivity by WB to pVEGFR2/VEGF and pERK/ERK – not shown), while in Kim et al. (1) the origin of the VEGF is not specified and in Wang et al. (2) the VEGF used comes from Carmeliet’s lab. It is possible then that the source of VEGF is critical to interpret these conflicting results. Without the missing information on nature (isoform) and source of VEGF in the papers by Wang and Kim, it will not be possible to test the reproducibility of their results. Critically, we find that despite the clear activation of VEGFR2 we see no evidence for activation of YAP or TAZ.

2) The VE-cadherin turnover needs to be better analysed. There is a body of evidence in the literature demonstrating that VE-cadherin internalization is key to increase vascular permeability in vitro and in vivo (Gavard and Gutkind, Nat. Cell Biol. 2006, 8:1223-1234; Gavard et al., Dev. Cell 2008, 14:25-36; Dwyer et al., Oncogene 2011, 30:190-200; Orsenigo et al., Nat. Commun. 2012, 3:1208; Yoshioka et al., Nat. Med. 2012, 18:1560-1569; Wessel, F. et al., Nat. Immunol. 2014, 15:223-230; Dorland et al., Nat. Commun. 2016, 7:12210; Di Russo et al., EMBO J. 2017, 36: 183-201). The observation that YAP/TAZ knockdown would increase endothelial cell permeability and at the same time reduce VE-cadherin endocytosis appears difficult to be reconciled in light of the above mentioned papers.The problem is that pulse-chase experiments provide only a semi-quantitative analysis of the VE-cadherin turnover at cell-cell contacts rather than a quantitative analysis VE-cadherin endocytosis or recycling. FRAP experiments may be a more precise quantitative approach to evaluate VE-cadherin turnover at cell-cell junctions in YAP/TAZ-depleted HUVECs. We also advise to characterise the colocalization of VE-cadherin with markers of early and/or late endosomes, such as EEA1 or LAMP1, respectively.

As suggested by the reviewers, we performed quantitative analysis of the VE-Cadherin turnover by fluorescence loss after photoconversion. When evaluating the best method to quantify VE-Cadherin turnover, we discovered that FRAP suffers from uncertainty when junctions move during the measurement. This happens frequently in endothelial cells. Therefore, we established VE-Cadherin tagged with photoconvertible mEos, allowing to quantify the loss of fluorescence after conversion of a subset of molecules, whilst monitoring location of the unconverted molecules. Using this method, we observed an increase in the immobile fraction of VE-Cadherin at the junction after YAP/TAZ knockdown (new Figure 5). This result corroborates and explains our previous observations in the pulse-chase experiments that showed decreased VE-Cadherin turnover.

As pointed out by the referees, our results showing increased permeability together with decreased endocytosis were surprising and conflicting with current concepts. To gain more insight into cause and potential mechanisms, we repeated the experiment as suggested, by doing co-staining with the endocytosis markers EEA1 and LAMP1.

To ensure that we studied internalised VE-Cadherin, we pulse labeled cells with the VE-cadherin 55/7H1 at 4C, allowed 30 min of internalization at 37C and performed an acid wash prior to fixing the cells to remove cell surface bound antibody. Acid wash treatment removed most of cell surface bound antibody, including junctional VE-Cadherin. In this baseline condition prior to challenging the junctions with calcium chelation, control and YAP/TAZ depleted cells showed no obvious differences in amount or localization of pulsed VE-cadherin staining (Author response image 2). Calcium chelation with EGTA led to dramatic internalization of VE-Cadherin in control cells, but very little after knockdown of YAP and TAZ, confirming our previous results.

**Author response image 2. respfig2:** Acid wash treatment removes cell surface bound antibody. HUVECs knocked down for YAP/TAZ (E-H) and control (A-D) were pulse labelled with VE-cadherin 55/7H1 for 30 mins at 4C and internalisation was allowed for 30 min at 37C in the presence of 5mM EGTA (B, D, F, H) or vehicle (A, C, E, G). Acid wash treatment (C, D, G, H) removes cell surface antibody revealing internalised VE-cadherin 55/7H1. Scale bar 50μm.

Before EGTA treatment, the early endosome marker EEA1 co-localised with the internalized VE-cadherin in both control and YAP/TAZ deficient cells (Author response image 3). However, we did not observe a co-localisation with LAMP1 (not shown). Interestingly, upon calcium chelation the dramatically internalized VE-cadherin did not appear in vesicles and did not co-localise with EEA1 (Author response image 3); instead, VE-cadherin was found as a belt around the nucleus, and in some cells with a filamentous pattern. These results suggest that the mechanism of internalization associated with the well-studied permeability process and that after calcium chelation are different processes.

**Author response image 3. respfig3:** Internalised VE-cadherin 55/7H1 co-localises with EEA1 in normal conditions, but not after disruption of cell junctions by calcium depletion. HUVECs knocked down for YAP/TAZ (B, D) and control (A, C) were pulse labelled with VE-cadherin 55/7H1-A647 (red) for 30 min at 4C and internalisation was allowed for 30 min at 37C in the presence of 5mM EGTA (C, D) or vehicle (A, B). Acid wash treatment was performed prior to fixing cells. Cells were co-stained with the early endosome marker EEA1 (green) and nuclei marker DAPI (blue). Yellow arrowheads, co-localisation of VE-cadherin 55/7H1-A647 with EEA1. Scale bar A-D, 10μm, A’-D’, 5μm.

Going back to the original discoveries of adherens junction dependence on calcium, we noticed that a similar pattern was observed in MDCK cells: upon calcium depletion adherens junction proteins together with actin filaments retract centripetally to a perinuclear position, and are not associated with membrane material (3-5). As such, instead of addressing VE-Cadherin endocytosis, these results instead suggest that in YAP/TAZ deficient cells adherens junctions are under less tension – either less actomyosin tension from the cytoskeleton or less linked to the actomyosin cytoskeleton. Interestingly, Kim et al. reported reduced phospho-myosin light chain 2 staining, indicating reduced tension in YAP/TAZ depleted endothelial cells (1). Our own f-actin staining showed strongly reduced branched actin networks and increased bundled actin at junctions in YAP/TAZ deficient cells, consistent with the observed changes in junctional patterning (new Figure 5). Together, these new results provide no evidence for altered endocytosis of VE-Cadherin, but confirm the reduced VE-cadherin turnover associated with altered junctional morphology in YAP/TAZ deficient endothelial cells. We propose it is this altered morphology together with reduced junction plasticity that is the cause for increased permeability.

3) We find the YAP/TAZ link interesting, although such a link has recently been proposed by Totaro et al. (2017). The role of Notch and its epistatic relationship with YAP and/or TAZ should be somehow tackled experimentally, at least in vitro, and demonstrated not in all but at least in few key experiments. There are good reagents to manipulate Notch in cell culture and to couple it as downstream effector of YAP/TAZ (at least partially). The authors should also consider to test whether Notch serves as a mechanical mediator of (some) YAP/TAZ biology for endothelial cells.

As we found Notch signaling to be repressed in cells with forced nuclear YAP or TAZ, and increased Notch signaling in YAP/TAZ deficient cells, we hypothesized that a Notch signaling increase could be responsible for the hyposprouting phenotype in *Yap*^fl/fl^
*Taz*^fl/fl^
*Pdgfb-iCreERT2.* However, the data from Kim et al. already argue against this hypothesis as treatment of YAP/TAZ endothelial mutant mice with DAPT (to inhibit Notch signalling) did not rescue the phenotype (1). Similarly, when testing the hypothesis further, we found that the in vitro treatment of YAP/TAZ deficient cells with DBZ was able to rescue the Notch signalling increase, but did not rescue the migration defect in the scratch wound assay (new Figure 8). The fact that recent literature links stronger Notch activity to reduced endothelial permeability (6), and yet we find increased permeability in YAP/TAZ loss of function conditions, also raises doubts whether the observed Notch increase is indeed a driver of the YAP/TAZ phenotype. Thus Notch signalling may not be the critical signalling pathway downstream of YAP/TAZ regulating behaviour in endothelial cells.

We then tested if the strong increase in BMP signalling in YAP/TAZ deficient cells was responsible for the phenotype. We found that a partial rescue in BMP signalling in vitroimproved the cell migration defect and corrected the permeability defect; it also rescued the increase in finger junctions in YAP/TAZ deficient cells. As we could not completely rescue the signalling deregulation in YAP/TAZ deficient cells, we can only conclude from our experiments that BMP signalling is at least partially responsible for the phenotype of YAP/TAZ loss. Our microarray data on YAP and TAZ gain of function mutant cells further showed that YAP and TAZ regulate the transcription of several BMP inhibitors, providing a plausible mechanistic link between YAP/TAZ and BMP signalling. The new data is included in Figure 8 and Figure 8—figure supplement 1. We have amended the discussion accordingly.

[Editors' note: further revisions were requested prior to acceptance, as described below.]

The manuscript has been improved but there are some remaining issues that need to be addressed before acceptance. In light of a recent paper published by the Schnittler lab in Nat. Commun., 2017, we would appreciate if you make the following changes in the text:

We have read and discussed the recent paper by the Schnittler lab in Nat. Comm. 2017 in depth in our team and we agree that it is an important piece of work, presenting data that relates to our own work and even supports our hypothesis. Specifically, Cao *et al.* were the first to identify the presence of junction associated intermediate lamellipodia (JAIL) in sprouting endothelial cells in vivo, thereby expanding the significance of these structures that to our knowledge had only been studied in cultured cells before, and so proving that their existence is not only an adaptation to the artificial in vitro culture system. Importantly, Cao *et al.* show that JAILs are present in sprouting endothelial cells (of the mouse retina and in the developing yolk sac), while they were not found in mature vessels. This strongly suggests that JAILs are characteristic of newly forming and highly dynamic angiogenic blood vessels. Therefore, the data from Cao *et al.* and our present work together would support a model in which decreased VE-Cadherin content at sprouting endothelial cells drives YAP and TAZ to the nucleus, where they increase the turnover of VE-Cadherin and increase the formation of JAILs, promoting cell migration and rearrangements. We have now included the reference Cao *et al.* 2017 and discuss the implications.

However, the very specific instructions of what to place in the Introduction of our manuscript and of adopting the terminology from Cao *et al.* throughout are problematic. After very careful consideration, we strongly feel that it is not appropriate or in the best interest of science to do so. In the following we explain our concerns and provide a point-by-point response:

1) Explain in the Introduction section that two classes of VE-cadherin-based adhesions exist: adherent junctions (characterizing endothelial cells of mature and already formed blood vessels) and JAILs (typical of endothelial cells of newly forming and very dynamic angiogenic blood vessels).

We don’t agree with this classification of VE-Cadherin based adhesions. Cell-cell junctions belong to one of three main categories: adherens junctions (AJ), tight junctions (TJ) and desmosomes. While in epithelial cells AJ and TJ are spatially separated, in the endothelium, with its characteristic flatness, this is not the case. In spite of that, AJ can be defined not only by (7) their spatial location in cell but also by (8) their molecular composition – the main components of adherens junctions are the cadherin molecules. Depending on the type of cell, AJ assume different morphologies: in polarised epithelial AJ is also called adhesion belt as it completely encloses the cells along with the F-actin lining; in fibroblasts AJ are discontinuous, and in neurons they are organized into tiny puncta as a constituent of the synaptic junctions (7). AJ in endothelial cells have been extensively studied and it was observed that there are different morphologies in endothelial cells – A) continuous/ linear/ straight junctions B), discontinuous/ fingers/serrated junctions and more recently also C) JAILs (8) (for illustration see Author response image 4). We therefore place JAIL as a subcategory of AJ, and not as a new category of VE-Cadherin based adhesions. These distinct morphological categories are associated with actin filaments in particular ways: continuous junctions overlap a bundled actin filament, discontinuous junctions are linked to actin filaments perpendicular to the junction border, and JAIL are linked to branched actin on the dependence of ARP2/3 activity. How these morphologies relate to the behaviour of the endothelial cells has been extensively investigated along with their description, but not often in vivo. Studies in the developing mouse retina vasculature have shown that continuous/ linear/ straight junctions were associated with stalk cell behaviour, while discontinuous/ fingers/serrated junctions were found in tip cells or actively rearranging cells (9). To this model, the new data from Cao *et al.* adds JAIL as a new category characteristic of sprouting, actively rearranging cells [added to the manuscript in line 210, highlighted in red]. A general new categorisation that sets JAIL apart from AJ would seem premature and we feel it is more correct to add JAIL as a new subcategory of AJ.

**Author response image 4. respfig4:** 

2) Adopt the Schnittler nomenclature throughout the manuscript by re-defining their "reticular junctions" as "large JAILs" and "thick to reticular junctions" as "small JAILs" (as also mentioned by authors in subsection “YAP/TAZ regulate adherens junction morphology and stability”). This will allow readers to more easily compare the complementary concepts exposed in these manuscripts.

An aspect that was raised already during this work and more so with this revision request it that the field of vascular biology will highly benefit from a consensus in nomenclature for adherens junctional categories. However, we believe it is important to most accurately describe the data in the Results section, without binning them into categories with assumed functions. For that reason, we defined the categories in morphological terms. Only subsequently, with live imaging, we were able to correspond thick to reticular and reticular junctions as small and large JAIL.

One important and novel aspect shown by Cao *et al.* by the use of live imaging is that these categories are not static. In fact, JAILs develop on top of continuous/ linear/ straight junctions and also on top of discontinuous/ fingers/ serrated junctions. In our work however, we characterized the morphology of AJ based on static data only. Because of this, we also caught what by live imaging data are clearly intermediate morphologies – as thick junctions and thick to reticular junctions.

In the paper from Cao *et al.,* endothelial cells are polarised by means of scratching a monolayer or by treatment with VEGF, so that they display lateral junctions and leading and trailing edge junctions. Cao *et al.* observed that, in polarised endothelial cells, lateral junctions associated with small JAIL and leading edge junctions associated with large JAIL. In our manuscript however, we did not analyse VE-Cadherin morphologies after polarising cells, so they do not display lateral and polar junctions. Our thick to reticular/small JAILs can be fully developed JAIL of smaller size or a developing or retracting JAIL or larger size. In this sense, adopting the Schnittlernomenclature would be overreaching the data – when we refer to small and large JAILs we refer to the size only, whereas they refer to a location in a polarized cell that is associated either with straight junctions or fingers (small JAIL – lateral junction – develops on top of straight junction and large JAIL – leading edge junction – develops on top of fingers). We propose that it is best to retain the strict morphology categories that can be directly connected to what the reader can see in the Results section. However, to embed the findings and provide functional interpretation and discussion, we refer to JAIL as discussion points.

3) Mention that the preferential nuclear YAP/TAZ localization in endothelial cells of the sprouting front of the developing retinal vasculature nicely fit with the observation reported by Cao et al. (Nat. Commun., 2017, 8:2210). Indeed, in Figure 1 Cao et al. show that endothelial cells of blood vessels localized at the sprouting front of the developing retina are elongated and display a reduced relative VE-cadherin concentration that promotes cell-to-cell junction dynamics and JAIL formation. Altogether, these data would provide additional strength to a model in which reduction of VE cadherin concentration at endothelial cell-to-cell junctions may promote YAP and TAZ relocation to the nucleus.

We have done so in the Discussion section, paragraph two.

1. Kim J, Kim YH, Kim J, Park DY, Bae H, Lee DH, Kim KH, Hong SP, Jang SP, Kubota Y, et al. YAP/TAZ regulates sprouting angiogenesis and vascular barrier maturation. *J Clin Invest.* 2017;127(9):3441-61.

2. Wang X, Freire Valls A, Schermann G, Shen Y, Moya IM, Castro L, Urban S, Solecki GM, Winkler F, Riedemann L, et al. YAP/TAZ Orchestrate VEGF Signaling during Developmental Angiogenesis. *Dev Cell.* 2017;42(5):462-78 e7.

3. Kartenbeck J, Schmid E, Franke WW, and Geiger B. Different modes of internalization of proteins associated with adhaerens junctions and desmosomes: experimental separation of lateral contacts induces endocytosis of desmosomal plaque material. *EMBO J.* 1982;1(6):725-32.

4. Volberg T, Geiger B, Kartenbeck J, and Franke WW. Changes in membrane-microfilament interaction in intercellular adherens junctions upon removal of extracellular Ca^2+^ ions. *J Cell Biol.* 1986;102(5):1832-42.

5. Kartenbeck J, Schmelz M, Franke WW, and Geiger B. Endocytosis of junctional cadherins in bovine kidney epithelial (MDBK) cells cultured in low Ca^2+^ ion medium. *J Cell Biol.* 1991;113(4):881-92.

6. Polacheck WJ, Kutys ML, Yang J, Eyckmans J, Wu Y, Vasavada H, Hirschi KK, and Chen CS. A non-canonical Notch complex regulates adherens junctions and vascular barrier function. *Nature.* 2017;552(7684):258-62.

7. Meng W, and Takeichi M. Adherens junction: molecular architecture and regulation. *Cold Spring Harb Perspect Biol.* 2009;1(6):a002899.

8. Abu Taha A, Taha M, Seebach J, and Schnittler HJ. ARP2/3-mediated junction-associated lamellipodia control VE-cadherin-based cell junction dynamics and maintain monolayer integrity. *Mol Biol Cell.* 2014;25(2):245-56.

9. Bentley K, Franco CA, Philippides A, Blanco R, Dierkes M, Gebala V, Stanchi F, Jones M, Aspalter IM, Cagna G, et al. The role of differential VE-cadherin dynamics in cell rearrangement during angiogenesis. *Nat Cell Biol.* 2014;16(4):309-21.

10. Cao J, Ehling M, Marz S, Seebach J, Tarbashevich K, Sixta T, Pitulescu ME, Werner AC, Flach B, Montanez E, et al. Polarized actin and VE-cadherin dynamics regulate junctional remodelling and cell migration during sprouting angiogenesis. *Nat Commun.* 2017;8(1):2210.